# Sparsely Changing Latent States for Prediction and Planning in Partially Observable Domains

**Christian Gumbsch**
Autonomous Learning Group
Max Planck Institute for Intelligent Systems
& Neuro-Cognitive Modeling Group
University of Tübingen
Tübingen, Germany
christian.gumbsch@tuebingen.mpg.de

**Martin V. Butz**
Neuro-Cognitive Modeling Group
University of Tübingen
Tübingen, Germany
martin.butz@uni-tuebingen.de

**Georg Martius**
Autonomous Learning Group
Max Planck Institute for Intelligent Systems
Tübingen, Germany
georg.martius@tuebingen.mpg.de

## Abstract

A common approach to prediction and planning in partially observable domains is to use recurrent neural networks (RNNs), which ideally develop and maintain a latent memory about hidden, task-relevant factors. We hypothesize that many of these hidden factors in the physical world are constant over time, changing only sparsely. To study this hypothesis, we propose Gated $L_0$ Regularized Dynamics (GateL0RD), a novel recurrent architecture that incorporates the inductive bias to maintain stable, sparsely changing latent states. The bias is implemented by means of a novel internal gating function and a penalty on the $L_0$ norm of latent state changes. We demonstrate that GateL0RD can compete with or outperform state-of-the-art RNNs in a variety of partially observable prediction and control tasks. GateL0RD tends to encode the underlying generative factors of the environment, ignores spurious temporal dependencies, and generalizes better, improving sampling efficiency and overall performance in model-based planning and reinforcement learning tasks. Moreover, we show that the developing latent states can be easily interpreted, which is a step towards better explainability in RNNs.

## 1 Introduction

When does the meeting start? Where are my car keys? Is the stove turned off? Humans memorize lots of information over extended periods of time. In contrast, classical planning methods assume that the state of the environment is fully observable at every time step [1]. This assumption does not hold for realistic applications, where generative processes are only indirectly observable or entities are occluded. Planning in such Partially Observable Markov Decision Processes (POMDP) is a challenging problem, because suitably-structured memory is required for decision making.

Recurrent neural networks (RNNs) are often used to deal with partial observability [2–4]. They encode past observations by maintaining latent states, which are iteratively updated. However, continuously updating the latent state causes past information to quickly "wash out". Long-Short Term Memory networks (LSTM, [5]) and Gated Recurrent Units (GRU, [6]) deal with this problem by using internal gates. However, they cannot leave their latent states completely unchanged, because

35th Conference on Neural Information Processing Systems (NeurIPS 2021).

small amounts of information continuously leak through the sigmoidal gating functions. Additionally, inputs typically need to pass through the latent state to affect the output, making it hard to disentangle observable from unobservable information within their latent states.

Our hypothesis is that many generative latent factors in the physical world are constant over extended periods of time. Thus, there might not be the need to update memory at every time step. For example, consider dropping an object: If the drop-off point as well as some latent generative factors, such as gravity and aerodynamic object properties, are known, iteratively predicting the fall can be reasonably accomplished by a non-recurrent process. Similarly, when an agent picks up a key, it is sufficient to memorize that the key is inside their pocket. However, latent factors typically do change significantly and systematically at particular points in time. For example, the aerodynamic properties of an object change drastically when the falling object shatters on the floor, and the location of the key changes systematically when the agent removes it from their pocket.

These observations are related to assumptions used in causality research. A common assumption is that the generative process of a system is composed of autonomous mechanisms that describe causal relationships between the system's variables [7–9]. When considering Markov Decision Processes, it has been proposed that these mechanisms tend to interact sparsely in time and locally in space [10, 11]. Causal models aim at creating dependencies between variables only when there exists a causal relationship between them, in order to improve generalization [8]. Updating the latent state of a model in every time step, on the other hand, induces the prior assumption that the generative latent state typically depends on all previous inputs. Thus, by suitably segmenting the dependencies of the latent variables over time, one can expect improved generalization across spurious temporal dependencies.

Very similar propositions have been made for human cognition. Humans tend to perceive their stream of sensory information in terms of events [12–16]. Event Segmentation Theory (EST) [16] postulates a set of active event models, which encode event-respective aspects over extended periods of time and switch individually at event transitions. To learn about the transitions and consolidate associated latent event encodings, measurements of surprise and other significant changes in predictive model activities, as well as latent state stability assumptions, have been proposed as suitable inductive event segmentation biases [16–22]. Explicit relations to causality have been put forward in [23].

In accordance to EST and our sparsely changing latent factor assumption, we introduce Gated $L_0$ Regularized Dynamics (GateL0RD). GateL0RD applies $L_0$-regularized gates, inducing an inductive learning bias to encode piecewise constant latent state dynamics. GateL0RD thus becomes able to memorize task-relevant information over long periods of time. The main contributions of this work can be summarized as follows. (i) We introduce a stochastic, rectified gating function for controlling latent state updates, which we regularize towards sparse updates using the $L_0$ norm. (ii) We demonstrate that our network performs as good or better than state-of-the-art RNNs for prediction or control in various partially-observable problems with piecewise constant dynamics. (iii) We also show that the inductive bias leads to better generalization under distributional shifts. (iv) Lastly, we show that the latent states can be easily interpreted by humans.

## 2 Background

Let $f_\theta : \mathcal{X} \times \mathcal{H} \to \mathcal{Y} \times \mathcal{H}$ be a recurrent neural network (RNN) with learnable parameters $\theta$ mapping inputs[1] $\boldsymbol{x}_t \in \mathcal{X}$ and $\boldsymbol{h}_{t-1} \in \mathcal{H}$ the latent (hidden) state to the output $\hat{\boldsymbol{y}}_t \in \mathcal{Y}$ and updated latent states $\boldsymbol{h}_t$. The training dataset $\mathcal{D}$ consists of sequences of input-output pairs $d = [(\boldsymbol{x}_1, \boldsymbol{y}_1), \ldots, (\boldsymbol{x}_T, \boldsymbol{y}_T)]$ of length $T$. In this paper, we consider the prediction and control of systems that can be described by a partially observable Markov decision process (POMDP) with state space $\mathcal{S}$, action space $\mathcal{A}$, observations space $\mathcal{O}$, and deterministic hidden transitions $\mathcal{S} \times \mathcal{A} \to \mathcal{S}$.[2]

---

[1]Notation: bold lowercase letters denote vectors (e.g., $\boldsymbol{x}$). Vector dimensions are denoted by superscript (e.g. $\boldsymbol{x} = [x^1, x^2, \ldots, x^n] \in \mathbb{R}^n$). Time or other additional information is denoted by subscript (e.g., $\boldsymbol{x}_t$).

[2]We treat the prediction of time series without any actions as a special case of the POMDP with $\mathcal{A} = \emptyset$.

# 3 $L_0$-regularization of latent state changes

We want the RNN $f_\theta$ to learn to solve a task, while maintaining piecewise constant latent states over time. The network creates a dynamics of latent states $h_t$ when applied to a sequence: $(\hat{y}_t, h_t) = f_\theta(x_t, h_{t-1})$ starting from some $h_0$. The most suitable measure to determine how much a time-series is piecewise constant is the $L_0$ norm applied to temporal changes. With the change in latent state as $\Delta h_t = h_{t-1} - h_t$, we define the $L_0$-loss as

$$\mathcal{L}_{L_0}(\Delta h) = \|\Delta h\|_0 = \sum_{j=1} \mathbb{I}(\Delta h^j \neq 0), \tag{1}$$

which penalizes the **number of non-zero entries** of the vector of latent state changes $\Delta h$.

The regularization loss from Eq. 1 can be combine in the usual way with the task objective to yield the overall learning objective $\mathcal{L}$ of the network:

$$\mathcal{L}(\mathcal{D}, \theta) = \mathbb{E}_{d \sim \mathcal{D}} \Big[ \sum_t \mathcal{L}_{\text{task}}(\hat{y}_t, y_t) + \lambda \mathcal{L}_{L_0}(\Delta h_t) \Big] \tag{2}$$

with $(\hat{y}_t, h_t) = f_\theta(x_t, h_{t-1})$. The task-dependent loss $\mathcal{L}_{\text{task}}(\cdot, \cdot)$ can be, for instance, the mean-squared error for regression or cross-entropy loss for classification. The hyperparameter $\lambda$ controls the trade-off between the task-based loss and the desired latent state regularization.

Unfortunately, we cannot directly minimize this loss using gradient-based techniques, such as stochastic gradient descent (SGD), due to the non-differentiability of the $L_0$-term. Louizos et al. [24] proposed a way to learn $L_0$ regularization of the learnable parameters of a neural network with SGD. They achieve this by using a set of stochastic gates controlling the parameters' usage. Each learnable parameter $\theta^j$ that is subject to the $L_0$ loss is substituted by a gated version $\theta'^j = \Theta(s^j)\theta^j$ where $\Theta(\cdot)$ is the Heaviside step function ($\Theta(s) = 0$ if $s \leq 0$ and 1 otherwise) and $s$ is determined by a distribution $q(s|\nu)$ with learned parameters $\nu$. Thus, $\theta'^j$ is only non-zero if $s^j > 0$. This allows to rewrite the $L_0$ loss (Eq. 1) for $\theta'$ as:

$$\mathcal{L}_{L_0}(\theta', \nu) = \|\theta'\|_0 = \sum_j \Theta(s^j) \qquad \text{with } s \sim q(s; \nu), \tag{3}$$

where parameters $\nu$ influence sparsity and are affected by the loss.

To tackle the problem of non-differentiable binary gates, we can use a smooth approximation as a surrogate [24–26]. Alternatively, we can substitute its gradients during the backward pass, for example using the straight-through estimator [27], which treats the step function as a linear function during the backward pass, or approximate its gradients as in the REINFORCE algorithm [28].

To transfer this approach to regularize the latent state dynamics in an RNN, we require an internal gating function $\Lambda(\cdot) \in [0, 1]$, which controls whether the latent state is updated or not. For instance:

$$h_t = h_{t-1} + \Lambda(s)\Delta \tilde{h}_{t-1} \qquad \text{with } \Delta \tilde{h}_{t-1} = \tilde{h}_t - h_{t-1} \tag{4}$$

where $\tilde{h}$ is the proposed new latent state and $s$ is a stochastic variable depending on the current input and previous latent state and the parameters, i.e. $s_t \sim q(s_t; x_t, h_{t-1}, \nu)$. For brevity, we merge the parameters $\nu$ into the overall parameter set, i.e. $\nu \subset \theta$. For computing Eq. 2 we need to binarize the gate by applying the step function $\Theta(\Lambda(s))$. Thus we can rewrite Eq. 2 as

$$\mathcal{L}(\mathcal{D}, \theta) = \mathbb{E}_{d \sim \mathcal{D}} \Big[ \sum_t \mathcal{L}_{\text{task}}(\hat{y}_t, y_t) + \lambda \sum_t \Theta(\Lambda(s_t)) \Big]. \tag{5}$$

LSTMs and GRUs use deterministic sigmoidal gates for $\Lambda$ in Eq. 4 to determine how to update their latent state. However, it is not straight forward to apply this approach to them (detailed in Suppl. A). Thus, we instead introduce a novel RNN, that merges components from GRUs and LSTMs, to implement the proposed $L_0$ regularization of latent state changes while still allowing the network to make powerful computations. We name our network Gated $L_0$ Regularized Dynamics (GateL0RD).

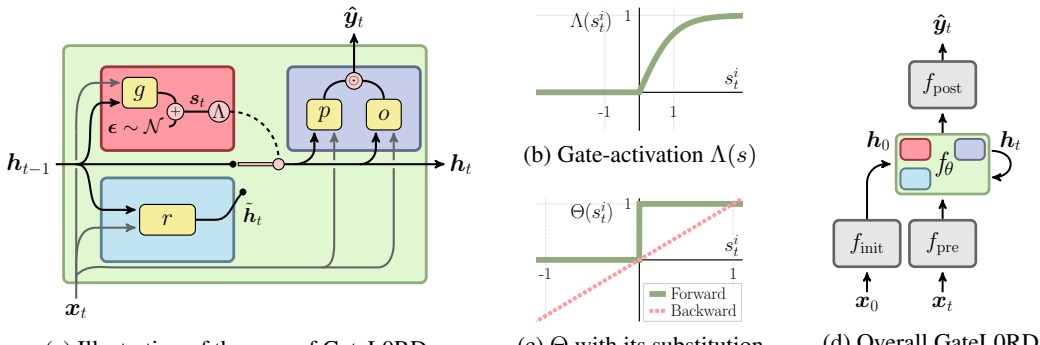

(a) Illustration of the core of GateL0RD.  (b) Gate-activation $\Lambda(s)$  (c) $\Theta$ with its substitution  (d) Overall GateL0RD

Figure 1: Architecture overview. (a) GateL0RD with its three subnetworks. The *gating function* controls the latent state update (red), the *recommendation function* computes a new latent state (blue) and the *output function* computes the output (purple). (b) Gate-activation function $\Lambda$ (ReTanh). (c) Heaviside step function $\Theta$ and its gradient estimator. (d) Overall architecture.

## 4   GateL0RD

The core of GateL0RD implements the general mapping $(\hat{\boldsymbol{y}}_t, \boldsymbol{h}_t) = f_\theta(\boldsymbol{x}_t, \boldsymbol{h}_{t-1})$ using three functions, or subnetworks: (1) a *recommendation network* $r$, which proposes a new candidate latent state, (2) a *gating network* $g$, which determines how the latent state is updated, and (3) an *output function*, which computes the output based on the updated latent state and the input. The network is systematically illustrated in Fig. 1a.

The overall processing is described by the following equations:

$$\boldsymbol{s_t} \sim \mathcal{N}(g(\boldsymbol{x}_t, \boldsymbol{h}_{t-1}), \boldsymbol{\Sigma}) \qquad \text{(sample gate input)} \qquad (6)$$

$$\Lambda(\boldsymbol{s}) := \max(0, \tanh(\boldsymbol{s})) \qquad \text{(new gating function)} \qquad (7)$$

$$\boldsymbol{h}_t = \boldsymbol{h}_{t-1} + \Lambda(\boldsymbol{s_t}) \odot (r(\boldsymbol{x}_t, \boldsymbol{h}_{t-1}) - \boldsymbol{h}_{t-1}) \qquad \text{(update or keep latent state)} \qquad (8)$$

$$\hat{\boldsymbol{y}}_t = p(\boldsymbol{x}_t, \boldsymbol{h}_t) \odot o(\boldsymbol{x}_t, \boldsymbol{h}_t), \qquad \text{(compute output)} \qquad (9)$$

where $\odot$ denotes element-wise multiplication (Hadamard product).

We start with the control of the latent state in Eq. 8. Following Eq. 4, a new latent value is proposed by the *recommendation function* $r(\boldsymbol{x}_t, \boldsymbol{h}_{t-1})$ and the update is "gated" by $\Lambda(\boldsymbol{s})$. Importantly, if $\Lambda(\boldsymbol{s}) = \boldsymbol{0}$ no change to the latent state occurs. Note that the update in Eq. 8 is in principle equivalent to the latent state update in GRUs [6], for which it is typically written as $\boldsymbol{h}_t = \Lambda(\boldsymbol{s}) \odot r(\boldsymbol{x}_t, \boldsymbol{h}_{t-1}) + (1 - \Lambda(\boldsymbol{s})) \odot \boldsymbol{h}_{t-1}$ with $\Lambda(\boldsymbol{s})$ a deterministic sigmoidal gate.

Because we aim for piecewise constant latent states, the gating function $\Lambda$ defined in Eq. 7 needs to be able to output exactly zero. A potential choice would be the Heaviside function, i.e. either copy the new latent state or keep the old one. This, however, does not allow any multiplicative computation. So a natural choice is to combine the standard sigmoid gate of RNNs with the step-function: $\Lambda(\boldsymbol{s}) = \max(0, \tanh(\boldsymbol{s}))$ which we call ReTanh (rectified tanh)[3]. Figure 1b shows the activation function $\Lambda$ depending on its input. The gate is closed ($\Lambda(s^i) = 0$) for all inputs $s^i \le 0$. A closed gate results in a latent state that remains constant in dimension $i$, i.e., $h_t^i = h_{t-1}^i$. On the other hand, for $s^i > 0$ the latent state is interpolated between the proposed new value and the old one.

The next puzzle piece is the input to the gate. Motivated from the $L_0$ regularization in Eq. 1 we use a stochastic input. However, in our RNN setting, it should depend on the current situation. Thus, we use a Gaussian distribution for $q$ with the mean determined by the *gating network* $g(\boldsymbol{x}_t, \boldsymbol{h}_{t-1})$ as defined in Eq. 6. We chose a fixed diagonal covariance matrix $\boldsymbol{\Sigma}$, which we set to $\Sigma^{i,i} = 0.1$. To train our network using backpropagation, we implement the sampling using the *reparametrization trick* [29]. We introduce a noise variable $\epsilon$ and compute the gate activation as

$$\boldsymbol{s}_t = g(\boldsymbol{x}_t, \boldsymbol{h}_{t-1}) + \boldsymbol{\epsilon} \qquad \text{with } \boldsymbol{\epsilon} \sim \mathcal{N}(\boldsymbol{0}, \boldsymbol{\Sigma}). \qquad (10)$$

During testing we set $\boldsymbol{\epsilon} = \boldsymbol{0}$ to achieve maximally accurate predictions.

---

[3]Note that $\tanh(s) = 2 \cdot \text{sigmoid}(2s) - 1$.

Finally the output $\hat{\boldsymbol{y}}$ is computed from the inputs and the new latent state $\boldsymbol{h}_t$ in Eq. 9. Inspired by LSTMs [5], the output is determined by a multiplication of a normal branch $(p(\boldsymbol{x}_t, \boldsymbol{h}_t))$ and a sigmoidal gating branch $(o(\boldsymbol{x}_t, \boldsymbol{h}_t))$. We thus enable both additive as well as multiplicative effects of $\boldsymbol{x}_t$ and $\boldsymbol{h}_t$ on the output, enhancing the expressive power of the piecewise constant latent states.

In our implementation, all subnetworks are MLPs. $r, p$ use a `tanh` output activation; $o$ uses a sigmoid; $g$ has a linear output. $p, o$ are one-layer networks. By default, $r, g$ are also one-layer networks. However, when comparing against deep (stacked) RNNs, we increase the number of layers of $r$ and $g$ to up to three (cf. Suppl. B).

We use the loss defined in Eq. 5. GateL0RD is fully differentiable except for the Heaviside step function $\Theta$ in Eq. 5. A simple approach to deal with discrete variables is to approximate the gradients by a differentiable estimator [25–27]. We employ the straight-through estimator [27], which substitutes the gradients of the step function $\Theta$ by the derivative of the linear function (see Fig. 1c).

We use GateL0RD as a memory module of a more general architecture illustrated in Fig. 1d. The network input is preprocessed by a feed-forward network $f_{\text{pre}}(\boldsymbol{x}_t)$. Similarly, its output is postprocessed by an MLP $f_{\text{post}}(\hat{\boldsymbol{y}}_t)$ (i.e. a readout layer) before computing the loss. The latent state $\boldsymbol{h}_0$ of GateL0RD could be initialized by $\boldsymbol{0}$. However, improvements can be achieved if the latent state is instead initialized by a context network $f_{\text{init}}$, a shallow MLP that sets $\boldsymbol{h}_0$ based on the first input [30, 31].

In the Supplementary Material we ablate various components of GateL0RD, such as the gate activation function $\Lambda$ (Suppl. C.1), the gate stochasticity (Suppl. C.2), the context network $f_{\text{init}}$ (Suppl. C.3), the multiplicative output branch $o$ (Suppl. C.4), and compare against $L_1/L_2$-variants (Suppl. C.5).

## 5 Related Work

**Structural regularization of latent updates:** Pioneering work on regularizing latent updates was done by Schmidhuber [32] who proposed the Neural History Compressor, a hierarchy of RNNs that autoregressively predict their next inputs. Thereby, the higher level RNN only becomes active and updates its latent states, if the lower level RNN fails to predict the next input. To structure latent state updates, the Clockwork RNN [33] partitions the hidden neurons of an RNN into separate modules, where each module operates at its own predefined frequency. Along similar lines, Phased LSTMs [34] use gates that open periodically. The update frequency in Clockwork RNNs and Phased LSTMs does not depend on the world state, but only on a predefined time scale.

**Loss-based regularization of latent updates:** For latent state regularization, Krueger and Memisevic [35] have proposed using an auxiliary loss term that punishes the change in $L_2$-norms of the latent state, which results in piecewise constant norms but not dynamics of the hidden states.

**Binarized update gates:** Closely related to our ReTanh, Skip RNNs [36] use a binary gate to determine latent state update decisions. Similarly, Gumbel-Gate LSTMs [37] replace sigmoid input and forget gates with stochastic, binary gates, approximated by a Gumbel-Softmax estimator [26]. Selective-Activation RNNs (SA-RNNs) [38] modify a GRU by masking the latent state with deterministic, binary gate and also incentivize sparsity. However, for GRUs the network output corresponds to the networks' latent state, thus, a piecewise constant latent state will result in piecewise constant outputs. All of these models were designed for classification or language processing tasks – none were applied for prediction or control in a POMDP setup, which we consider here.

**Attention-based latent state updates:** Sparse latent state updates can also be achieved using attention [39–41]. Neural Turing Machines [39] use an attention mechanism to update an external memory block. Thereby, the attention mechanism can focus and only modify a particular locations within the memory. Recurrent Independent Mechanisms (RIMs) [42] use a set of recurrent cells that only sparsely interact with the environment and one another through competition and a bottleneck of attention. Recent extensions explore the update of the cells and the attention parameters at different time scales [43]. For RIMs the sparsity of the latent state changes is predefined via a hyperparameter that sets the number of active cells. In contrast, our $L_0$ loss implements a soft constraint.

**Transformers:** Transformers [41] omit memory altogether, processing a complete sequence for every output at once using key-based attention. While this avoids problems arising from maintaining a latent state, their self-attention mechanism comes with high computational costs. Transformers

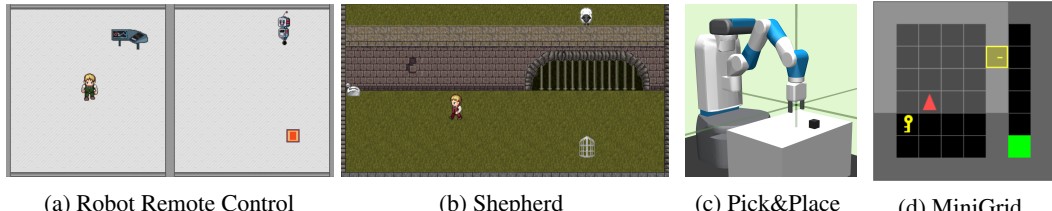

| (a) Robot Remote Control | (b) Shepherd | (c) Pick&Place | (d) MiniGrid |

Figure 2: Simulations used to test GateL0RD. (a) and (b) are continuous 2D-control tasks: (a) requires triggering the control of a robot by getting a remote control; (b) needs memorization of the sheep's position to capture it later. (c) is the Fetch Pick&Place environment [47] modified to become partially observable and (d) shows a problem (DoorKey-8x8) of the Mini-Gridworld suite [48].

have shown breakthrough success in natural language processing. However, it remains challenging to train them for planning or reinforcement learning applications in partially-observable domains [44].

## 6 Experiments

Our experiments offer answers to the following questions: (a) Does GateL0RD generalize better to out-of-distribution inputs in partially observable domains than other commonly used RNNs? (b) Is GateL0RD suitable for control problems that require (long-term) memorization of information? (c) Are the developing latent states in GateL0RD easily interpretable by humans? Accordingly, we demonstrate both GateL0RD's ability to generalize from a 1-step prediction regime to autoregressive $N$-step prediction (Sec. 6.1) and its prediction robustness when facing action rollouts from different policies (Sec. 6.2). We then reveal precise memorization abilities (Sec. 6.3) and show that GateL0RD is more sample efficient in various decision-making problems requiring memory (Sec. 6.4). Finally, we examine exemplary latent state codes demonstrating their explainability (Sec. 6.5).

In our experiments we compare GateL0RD to LSTMs [5], GRUs [6], and Elman RNNs [45]. We use the architecture shown in Fig. 1d for all networks, only replacing the core $f_\theta$. We examine the RNNs both as a model for *model-predictive control* (MPC) as well as a memory module in a *reinforcement learning* (RL) setup. When used for prediction, the networks received the input $x_t = (o_t, a_t)$ with observations $o_t \in \mathcal{O}$ and actions $a_t \in \mathcal{A}$ at time $t$ and were trained to predict the change in observation, i.e. $y_t = \Delta o_{t+1}$ (detailed in Suppl. B.1). During testing the next observational inputs were generated autoregressively as $\hat{o}_{t+1} = o_t + \hat{y}_t$. In the RL setting, the networks received as an input $x_t = o_t$ the observation $o_t \in \mathcal{O}$ and were trained as an actor-critic architecture to produce both policy and value estimations (detailed in Suppl. B.6). The networks were trained using Adam [46], with learning rates and layer numbers determined via grid search for each network type individually (cf. Suppl. B).

We evaluate GateL0RD in a variety of partially observable scenarios. In the **Billiard Ball** scenario a single ball, simulated in a realistic physics simulator, is shot on a pool table with low friction from a random position in a random direction with randomly selected velocity. The time series contain only the positions of the ball. This is the only considered scenario without actions.

**Robot Remote Control** is a continuous control problem where an agent moves according to the two-dimensional actions $a_t$ (Fig. 2a). Once the agent reaches a fixed position (terminal), a robot in another room is also controlled by the actions. The observable state $o_t$ is composed of the agent's position and the robot's position. Thus, whether the robot is controlled or not is not observable directly. When planning, the goal is to move the robot to a particular goal position (orange square).

**Shepherd** is a challenging continuous control problem that requires long-term memorization (Fig. 2b). Here, the agent's actions $a_t$ are the two movement directions and a grasp action controlling whether to pick up or drop the cage. The sheep starts at the top of the scene moving downwards with a fixed randomly generated velocity. The sheep is then occluded by the wall, which masks its position from the observation. If the agent reaches the lever, the gate inside the wall opens, and the sheep appears again at the same horizontal position at the open gate. The goal is to get the sheep to enter the previously placed cage. The challenge is to memorize the sheep's horizontal position exactly over a potentially long time to place the cage properly and to then activate the lever during mental

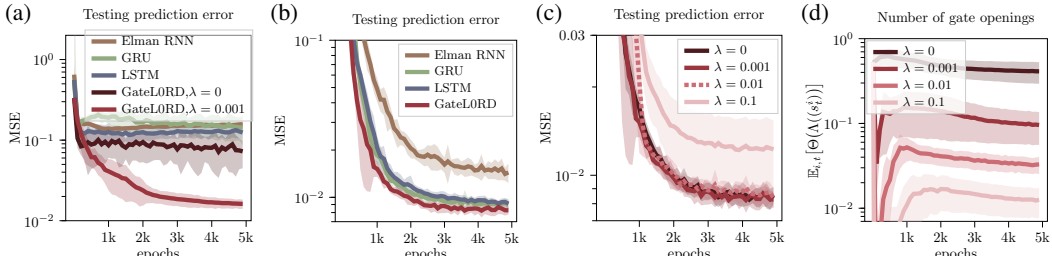

Figure 3: Billiard Ball results: prediction errors when trained using teacher forcing (a), or using scheduled sampling (b). GateL0RD's prediction error (c) and mean number of gate openings (latent state updates) (d) for different values of $\lambda$. Shaded areas show $\pm$ one standard deviation.

simulation. The seven-dimensional observation $o_t$ is composed of the height of the occluder and the positions of all entities.

**Fetch Pick&Place** (OpenAI Gym v1, [47]) is a benchmark RL task where a robotic manipulator has to move a randomly placed box (Fig. 2c). In our modified setting[4], the observable state $o_t$ is composed of the gripper- and object position and the relative positions of object and fingers with respect to the gripper. The four-dimensional actions $a_t$ control the gripper position and the opening of the fingers.

**MiniGrid** [48] is a gridworld suite with a variety of partially observable RL problems. At every time $t$, the agent (red triangle in Fig. 2d) receives an image-like, restricted, ego-centric view (grey area) as its observation $o_t$ ($7 \times 7 \times 3$-dimensional). It can either move forward, turn left, turn right, or interact with objects via its one-hot-encoded actions $a_t$. The problems vary largely in their difficulty, typically contain only sparse rewards, and often involve memorization, e.g., remembering that the agent picked up a key. Suppl. B.7 details all examined MiniGrid environments.

## 6.1 Learning autoregressive predictions

First, we consider the problem of autoregressive $N$-step prediction in the **Billiard Ball** scenario. Here, during testing the networks receive the first two ball positions as input and predict a sequence of 50 ball positions. We first train the RNNs using *teacher forcing*, whereby the real inputs are fed to the networks. Figure 3a shows the prediction error for autoregressive predictions. Only GateL0RD with latent state regularization ($\lambda = 0.001$) is able to achieve reasonable predictions in this setup. The other RNNs seem to learn to continuously update their estimates of the ball's velocity based on the real inputs. Because GateL0RD punishes continuous latent state updates, learning leads to updates of the estimated velocity only when required, i.e. upon collisions, improving its prediction robustness.

The problems of RNNs learning autoregressive prediction are well known [49, 50]. A simple countermeasure is *scheduled sampling* [49], where each input is stochastically determined to be either the last network's output or the real input. The probability of using the network output increases over time. While the prediction accuracy of all RNNs improves when trained using scheduled sampling, GateL0RD ($\lambda = 0.001$) still achieves the lowest mean prediction error (see Fig. 3b).

How does the regularization affect GateL0RD? Figure 3c shows the prediction error for GateL0RD for different settings of $\lambda$. While a small regularization ($\lambda = 0.001$) leads to the highest accuracy in this scenario, similar predictions are obtained for different strengths ($\lambda \in [0, 0.01]$). Overly strong regularization ($\lambda = 0.1$) degrades performance. Figure 3d shows the average gate openings per sequence. As indented, $\lambda$ directly affects how often GateL0RD's latent state is updated: a higher value results in fewer gate openings and, thus, fewer latent state changes. Note that even for $\lambda = 0$ GateL0RD learns to use fewer gates over time. We describe this effect in more detail in Suppl. D.1.

## 6.2 Generalization across policies

Particularly when priorities change or an agent switches behavior, different spurious temporal correlations can occur in the resulting sensorimotor timeseries data. Consequently, models are needed

---

[4]We omit all velocities and the rotation of the object to make the scenario partially observable.

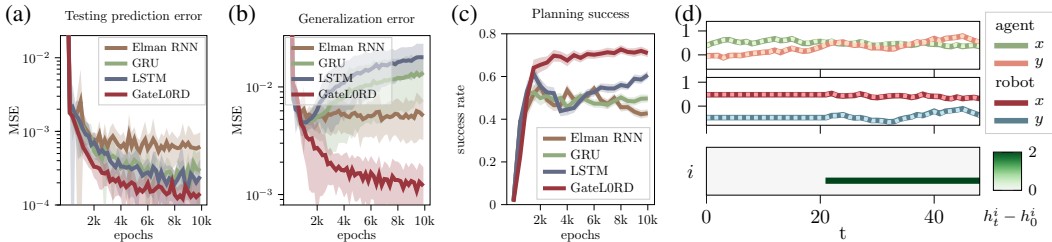

Figure 4: Robot Remote Control results: prediction error on the test set (a) and on the generalization set (b). Success rate for MPC (c). Shaded areas show standard deviation (a & b) or standard error (c). Exemplary generalization sequence (d) showing the agent's positions (top), the robot's positions (middle) with GateL0RD's predictions shown as dots, and GateL0RD's latent states (bottom).

that generalize across those correlations. We use the networks trained as predictive models for the **Robot Remote Control** scenario to investigate this aspect.

In Robot Remote Control the training data is generated by performing rollouts with 50 time steps of a policy that produces random but linearly magnitude-increasing actions. The actions' magnitude in the training data is positively correlated with time, which is a spurious correlation that does not alter the underlying transition function of the environment in any way. We train the networks to predict the sequence of observations given the initial observation and a sequence of actions. Thereby, we test the networks using data generated by the same policy (*test set*) and generated by a policy that samples uniformly random actions (*generalization set*). Additionally, we use the trained RNNs for model-predictive control (MPC) using iCEM [51], a random shooting method that iteratively optimizes its actions to move the robot to the given goal position.

As shown in Fig. 4a, GateL0RD ($\lambda = 0.001$) outperforms all other RNNs on the test set. When tested on the generalization data, the prediction errors of the GRU and LSTM networks even increase over the course of training. Only GateL0RD is able to maintain a low prediction error. Figure 4c shows the MPC performance. GateL0RD yields the highest success rate.

Note that the lack of generalization is not primarily caused by the choice of hyperparameters: even when the learning rate of the other RNNs was optimized for the generalization set, GateL0RD still outperformed them (additional experiment in Suppl. D.3). Instead, GateL0RD's better performance is likely because it mostly encodes unobservable information within its latent state $h_t$. This is shown exemplarily in Fig. 4d (bottom row) and analyzed further in Suppl. D.5. The latent state remains constant and only one dimension changes once the agent controls the robot's position (middle row) through its actions. Because the other RNNs also encode observable information, e.g. actions, within their latent state, they are more negatively affected by distributional shifts and spurious dependencies.

GateL0RD's improved generalization across temporal dependencies also holds for more complicated environments. In an additional experiment in Suppl. D.7 we show similar effects for the **Fetch Pick&Place** environment when trained on reach-grasp-and-transport sequences and tested to generalize across timings of the grasp.

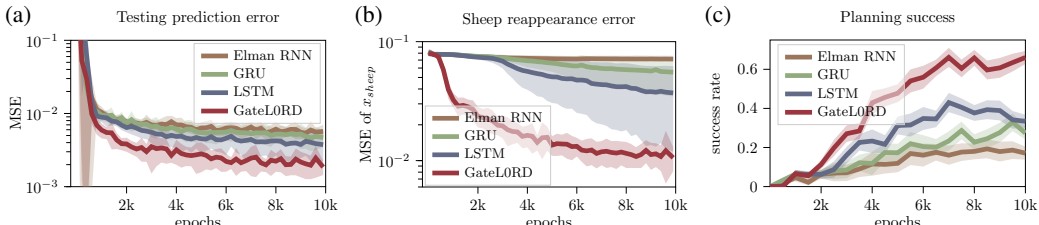

Figure 5: Shepherd results: prediction error for 100-step predictions (a) and 1-step prediction errors of the sheep's $x-$position at the time step of reappearance (b). Success rate for capturing the sheep using MPC (c). Shaded areas show standard deviation (a-b) or standard error (c).

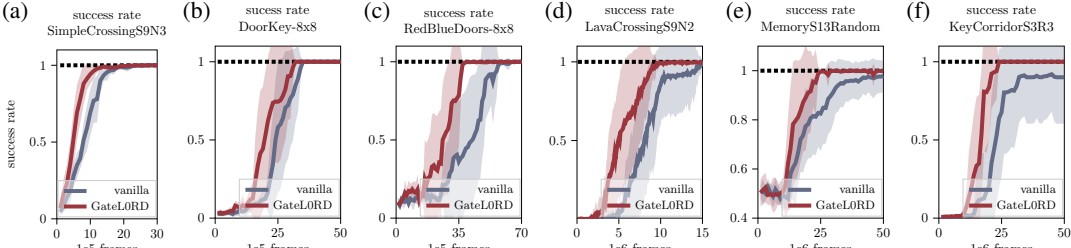

Figure 6: MiniGrid results: success rate in solving various tasks when GateL0RD replaces an LSTM (vanilla) in a PPO architecture. Shaded areas depict the standard deviation.

## 6.3 Long-term memorization

We hypothesized that GateL0RD's latent state update strategy fosters the exact memorization of unobservable information, which we examine in the **Shepherd** task. We test the RNNs' when predicting sequences of 100 observations given the first two observations and a sequence of actions. Again, we use the trained models for MPC using iCEM [51], aiming at catching the sheep by first placing a cage and then pulling a lever. This is particularly challenging to plan because the sheep's horizontal position needs to be memorized before it is occluded for quite some time ($> 30$ steps) in order to accurately predict and thus place the cage at the sheep's future position.

Figure 5a shows the prediction errors during training. GateL0RD ($\lambda = 0.0001$) continuously achieves a lower prediction error than the other networks. Apparently, it is able to accurately memorize the sheep's future position while occluded. To investigate the memorization we consider the situation occurring during planning: the sequence of (past) observations is fed into the network and the prediction error of the sheep's horizontal position at the time of reappearance is evaluated (Fig. 5b). Only GateL0RD reliably learns to predict where the sheep will appear when the lever is activated. GRU and Elman RNNs do not noticeably improve in predicting the sheep's position. LSTMs take much longer to improve their predictions and do not reliably reach GateL0RD's level of accuracy. This is also reflected in the success rate when the networks are used for MPC (Fig. 5c). Only GateL0RD manages to solve this challenging task with a mean success rate over 50%.

## 6.4 Sample efficiency in reinforcement learning

Now that we have outlined some of GateL0RD's strengths in isolation, we want to analyze whether GateL0RD can improve existing RL-frameworks when it is used as a memory module for POMDPs. To do so, we consider various problems that require memory in the **MiniGrid** suite [48]. Previous work [42, 43, 52] used Proximal Policy Optimization (PPO) [53] to solve the MiniGrid problems. We took an existing architecture based on [52] (denoted as *vanilla*, detailed in Suppl. B.6) and replaced the internal LSTM module with GateL0RD ($\lambda = 0.01$). Note, that we left the other hyperparameters unmodified.

As shown in Fig. 6 the architecture containing GateL0RD achieves the same success rate or higher than the vanilla baseline in all considered tasks. Additionally, GateL0RD is more sample efficient, i.e., it is able to reach a high success rate (Fig. 6) or high reward level faster (Suppl. D.9). The difference in sample efficiency tends to be more pronounced for problems that require more training time. It seems that the inductive bias of sparsely changing latent states enables GateL0RD to quicker learn to encode task-relevant information, such as the pick-up of a key, within its latent states. Additional experiments in Suppl. D.10 show that this can also translates to improved zero-shot policy transfer, when the system is tested on a larger environment than it was trained on.

## 6.5 Explainability of the latent states

Lastly, we analyze the latent representations of GateL0RD, starting with **Billiard Ball**. Figure 7a shows one exemplary ball trajectory in white and the prediction in red. Inputs for which at least one gate opened are outlined in black. Figure 7b shows the corresponding latent states $h_t$ relative to the initial latent state $h_0$. GateL0RD updates two dimensions of its latent states around the points of collisions to account for the changes in $x$- and $y$-velocity of the ball. For $\lambda = 0.01$ we find on

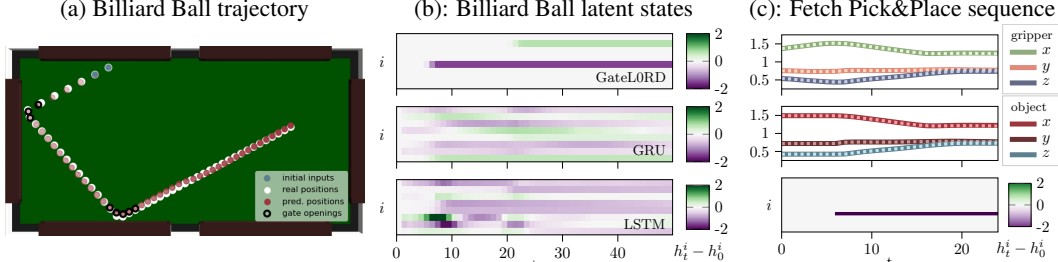

Figure 7: Example sequences and latent states: (a) Billiard Ball trajectory for GateL0RD ($\lambda = 0.01$) with real positions (white), provided inputs (blue), and predicted positions (red, saturation increasing with time). The inputs for which at least one gate opened are outlined in black. (b) The latent states for the trajectory for GateL0RD, GRU, and LSTM (cell states). (c) Fetch Pick&Place sequence with real (solid) and predicted (dotted) positions of gripper (top) and object (middle) and GateL0RD's latent states (bottom). Latent states are shown relative to initialization, i.e. $\boldsymbol{h}_t - \boldsymbol{h}_0$.

average only two latent state dimensions change per sequence (see Suppl Suppl. D.1), which hints at a tendency to encode $x$- and $y$-velocity using separate latent dimensions. In contrast, the exemplary latent states of the GRU and LSTM networks shown in Fig. 7b are not as easily interpretable.

For **Robot Remote Control**, GateL0RD ($\lambda = 0.001$) updates only its latent state once it controls the robot (exemplary shown in Fig. 4d). Thus, the latent state clearly encodes control over the robot. We use the **Fetch Pick&Place** scenario as a higher-dimensional problem to investigate latent state explainability when training on grasping sequences (detailed in Suppl. B.5). Here, GateL0RD updates the latent state typically when the object is grasped (exemplary shown in Fig. 7c). This hints at an encoding of 'object transportation' using one dimension. Other RNNs do not achieve such a clear representation, neither in Robot Remote Control nor in Fetch Pick&Place (see Suppl. D.5 and D.7).

## 7   Discussion

We have introduced a novel RNN architecture (GateL0RD), which implements an inductive bias to develop sparsely changing latent states. The bias is realized by a gating mechanism, which minimizes the $L_0$ norm of latent updates. In several empirical evaluations, we quantified and analyzed the performance of GateL0RD on various prediction and control tasks, which naturally contain piecewise constant, unobservable states. The results support our hypothesis that networks with piecewise constant latent states can generalize better to distributional shifts of the inputs, ignore spurious time dependencies, and enable precise memorization. This translates into improved performance for both model-predictive control (MPC) and reinforcement learning (RL). Moreover, we demonstrated that the latent space becomes interpretable, which is important for explainability reasons.

Our approach introduces an additional hyperparameter, which controls the trade-off between the task at hand and latent space constancy. When chosen in favor of explainability, it can reduce the in-distribution performance while improving its generalization abilities. When the underlying system has continuously changing latent states, our regularization is counterproductive. As demonstrated by an additional experiment in Suppl. D.8, the unregularized network performs well in such cases.

Our sparsity-biased gating mechanism segments sequences into chunks of constant latent activation. These segments tend to encode unobservable, behavior-relevant states of the environment, such as if an object is currently 'under control'. Hierarchical planning and control methods require suitable, temporally-extended encodings, such as options [54, 55]. Thus, a promising direction for future work is to exploit the discrete hidden dynamics of GateL0RD for hierarchical, event-predictive planning.

### Acknowledgments and Disclosure of Funding

The authors thank the International Max Planck Research School for Intelligent Systems (IMPRS-IS) for supporting Christian Gumbsch. Georg Martius and Martin Butz are members of the Machine Learning Cluster of Excellence, EXC number 2064/1–project number 390727645. We acknowledge

the support from the German Federal Ministry of Education and Research through the Tübingen AI Center (FKZ: 01IS18039B). This research was funded by the German Research Foundation (DFG) within Priority-Program "The Active Self" SPP 2134–project BU 1335/11-1. The authors thank Maximilian Seitzer for the helpful feedback and Sebastian Blaes for the help in applying iCEM.

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
