# Supplementary Material for:
# Sparsely Changing Latent States for Prediction and Planning in Partially Observable Domains

## A    Relation to other RNNs

In Sec. 3 we set out to create an RNN $f_\theta$ that maintains piecewise constant latent states over time. This led us to the conclusion, that a simple approach to implement this is by employing an internal gating function $\Lambda$ that controls the latent state update (e.g. as in Eq. 4). The gating function $\Lambda$ can be binarized using the Heaviside step function and a sparse gating can be incentivized using the loss function outlined in Eq. 5.

GRUs [6] and LSTMs [5] both use internal gates with the sigmoid activation function $\sigma$ to control the update of their latent state $h_t$. GRUs update $h_t$ with

$$h_t = (1 - \sigma(s_t))h_{t-1} + \sigma(s_t)\tilde{h}_t, \qquad (11)$$

where $s_t$ is a linear projection of input $x_t$ and previous latent state $h_{t-1}$ and $\tilde{h}_t$ is a proposed new latent state, also determined based on the input $x_t$ and previous latent state $h_{t-1}$.

LSTMs use two gates, i.e. a forget and an input gate, with the sigmoid activation function $\sigma$ to determine whether to update their latent (cell) state $h_t$ with

$$h_t = \sigma(s_{1,t})h_{t-1} + \sigma(s_{2,t})\tilde{h}_t, \qquad (12)$$

where $s_{1,t}$ and $s_{2,t}$ are linear projections and $\tilde{h}_t$ a non-linear function of the input and previous hidden state (RNN cell output).

Nonetheless, it is not straight forward to apply our approach, outlined in Sec. 3, to GRUs and LSTMs. Our loss (see Eq. 5) punishes non-zero gate activation. The sigmoid activation function $\sigma$ only achieves an output of zero if its input converges to negative infinity, thus, never truly achieving zero output. Thus, their gating function would need to be modified or replaced, e.g. by our ReTanh gate $\Lambda$.

However, even when replacing their gate activation function, the performance of LSTMs and GRUs are negatively affected by piecewise constant latent states. For both networks, input information essentially needs to pass through the latent state to affect the network output. For GRUs the network output corresponds to the latent state $h_t$. Thus, a GRU with constant latent states will produce constant outputs. In LSTMs the network output is computed by multiplying the latent (cell) state with an input-dependent output gate. Thus, in LSTMs a constant latent state will result in a constant output that is scaled depending on the network input.

GateL0RD attempts to overcome the outlined downsides of using LSTMs and GRUs with our proposed latent state regularization. Like GRUs, GateL0RD uses a single update gate to avoid unnecessary parameters. Additionally, GateL0RD separates the latent state from the network output, as done in LSTMs, which have both a cell state and a hidden state. Besides that, GateL0RD uses more powerful functions for computing the network output such that input and latent state both have an additive as well as multiplicative effects on the network output. Note that GateL0RD still has approximately the same number of parameters as a GRU.

## B    Experimental Details

### B.1    Predictive Models: General Training Principles and Hyperparameter Search

In the following, we will outline the general training principles that we used in all experiments, when the RNNs were trained as *predictive models*. Training details for the reinforcement learning experiments are found in Suppl. B.6. Suppl. B.2 - B.5 provide further details specific to each simulation independent of the hyperparameter search (e.g. dataset size, batch size, etc.).

In our experiments, we train each network to predict the change in observations instead of the next observation (i.e. residual connections) to avoid the trivial solution of achieving a high prediction accuracy by simply outputting the input observation. However, since the change in observation can

Table 1: Learning rate choices

| Experiment | GateL0RD | LSTM | GRU | Elman RNN |
|---|---|---|---|---|
| **Billiard Ball** teacher forcing (Sec. 6.1) | 0.001 | 0.001 | 0.001 | 0.00005 |
| **Billiard Ball** scheduled sampling (Sec. 6.1) | 0.0005 | 0.0005 | 0.0005 | 0.0005 |
| **Robot Remote Control (RRC)** (Sec. 6.2) | 0.005 | 0.005 | 0.005 | 0.005 |
| **RRC** improved generalization (Suppl. D.3) | 0.005 | 0.001 | 0.001 | 0.001 |
| **Shepherd** (Sec. 6.3) | 0.001 | 0.001 | 0.001 | 0.001 |
| **Fetch Pick&Place** filtered data (Suppl. D.7) | 0.005 | 0.005 | 0.005 | 0.005 |
| **Fetch Pick&Place** full data (Suppl. D.8) | 0.001 | 0.001 | 0.001 | 0.001 |

be quite small (typically $\Delta o_t < 0.1$) we use a constant $c$ to scale the network output when used as autoregressive input, i.e. $\hat{o}_{t+1} = o_t + c \cdot \hat{y}_t$. We set $c = 0.1$ in all our experiments, which corresponds to scaling $\Delta o_t$ by a factor of 10. For the task-based loss, i.e. $\mathcal{L}_{\text{task}}$ in Eq. 2, we use the mean squared error between predicted observations $\hat{o}_t$ and real observations $o_t$.

We train the networks using Adam [46] with the hyperparameters $\beta_1 = 0.9$, $\beta_2 = 0.999$, and $\epsilon = 0.0001$. The learning rate $\alpha$ was determined via a grid search with $\alpha \in \{0.005, 0.001, 0.0005, 0.0001, 0.00005\}$ for each scenario. For this grid search, we examined two random seeds for each parameter configuration and chose the setting resulting in the lowest mean squared prediction error on a validation set after full training. The best learning rates for all experiments are listed in Table 1.

Besides determining the learning rate, we also use grid search to determine the number of RNN layers for all scenarios with simulated physics, i.e. Billiard Ball and Fetch Pick&Place. For LSTMs, GRUs, and Elman RNNs we compare the 1-layered RNNs to a stacked version in which up to three RNN cells ($f_\theta$ in Fig. 1d) are composed. For GateL0RD we instead considered 1- to 3-layered $r$ and $g$-networks (see Fig. 1a), since we found that this typically results in a stronger increase in performance with fewer parameters compared to stacking GateL0RD cells. In Billiard Ball (Sec. 6.1) and Fetch Pick&Place (full data, Suppl. D.8), all networks achieve a slightly better mean prediction accuracy with the 3-layered versions, which is why we use the 3-layered versions to compare the prediction accuracy. However, for GRUs and LSTMs the 3-layered versions have three times the number of latent state dimensions, which negatively affects the interpretability of the latent states. Thus, to make a fair comparison in terms of explainability, we additionally ran experiments with 1-layered LSTMs and GRUs to visualize the latent states (e.g. in Fig. 7b). For Fetch Pick&Place with pre-selected reach-grasp-lift sequences (Suppl. D.7) there was no noticeable improvement when increasing the number of layers, thus, we used one-layered versions of the networks.

RNNs can suffer from the exploding gradient problem when predicting long sequences. An effective technique to deal with this is *gradient norm clipping* [56]. Here, the norm of a backpropagated gradient is clipped when it exceeds a threshold. We apply gradient norm clipping in all our experiments with a clipping threshold of 0.1.

In Sec. 6.1 we showed that training the models using teacher forcing can be problematic. Thus, in all of our other experiments, we train the networks using scheduled sampling [49], a curriculum learning strategy that smoothly changes the training regime from teacher forcing to autoregressive predictions. When applying scheduled sampling, a probability $p_i$ is used to stochastically determine whether the real input is fed into the network (teacher forcing) or whether to use the previous network output. This sampling probability $p_i$ decreases over training time $i$. Based on Bengio et al. [49], we use an exponentially decreasing probability $p_i$ with

$$p_i = \max(k^i, p_{\min}) \tag{13}$$

where $i$ is the epoch number, $k < 1$ a constant, and $p_{\min}$ the minimum sampling probability. We set $k = 0.998$ in all experiments. The minimum sampling probability $p_{\min}$ is chosen individually for each scenario.

All experiments using predictive models were run with 20 different random seeds for each setting.

## B.2 Billiard Ball

In the Billiard Ball scenario, a ball is shot on a pool table with low friction. We generated sequences of 50 time steps by shooting the ball from a random starting position in a random direction with

a randomly selected velocity. The sequences were generated using the Open Dynamics Engine (ODE)[5]—an open-source physics simulator for simulating rigid-body dynamics. The sequences contain only the observations $o_t \in [-1, 1]^2$, which are composed of the positions of the ball, and no actions ($\mathcal{A} = \emptyset$).

The networks were trained on a training set of 12.8k sequences and tested on a testing set of 3.2k sequences. Hyperparameters were determined based on a validation set of 3.2k sequences. All datasets were balanced to include different velocities and to guarantee that in at least $15\%$ of the sequences the ball drops into a pocket. We trained the networks using minibatches of size 128 for 5k epochs. We applied scheduled sampling [49] by exponentially annealing the sampling probability $p_i$ to 0.

We used an 8-dimensional latent state $h_t$ for all RNNs. The latent state $h_0$ was initialized based on the first two inputs using a 3-layered MLP $f_{\text{init}}$ (neurons per layer: $64 \rightarrow 32 \rightarrow 16$). All RNNs used a 3-layered MLP $f_{\text{pre}}$ (neurons per layer: $64 \rightarrow 32 \rightarrow 16$) for preprocessing the inputs and a single linear mapping as a readout layer $f_{\text{post}}$.

### B.3  Robot Remote Control

In the Robot Remote Control scenario, an agent continuously moves through a room based on its two-dimensional actions $a_t \in [-1, 1]^2$. After the agent reaches a computer, it also controls the position of a robot in another room through its actions. The goal during planning is to move the robot to a goal area. The observation $o_t \in [-1, 1]^4$ is composed of the position of the agent and the position of the robot. The robot and agent start from randomly sampled positions while the computer and goal area are always at the same fixed positions. The robot is controlled as soon as the distance between agent and computer is below a certain interaction threshold (0.1).

We generated datasets composed of 50 time step rollouts using two synthetic policies. The dataset $\mathcal{D}_{\text{time}}$, containing spurious temporal dependencies, was generated by sampling uniformly distributed random actions that were scaled by a factor that linearly increases with time from 0.0001 to 1.0. The (generalization) dataset $\mathcal{D}_{\text{random}}$ was generated by sampling uniformly distributed random actions without further modifications. Both datasets were balanced in terms of robot control events, such that in half of the sequences the robot was controlled by the agent. The datasets were split into equally sized training, validation, and testing sets (6.4k sequences each). The validation sets were used to determine hyperparameters. The networks were trained for 5k epochs using minibatches of size 128. We trained the networks using scheduled sampling [49] by exponentially annealing the sampling probability $p_i$ to a minimum value of $p_{\text{min}} = 0.02$.

In this scenario, the latent states $h_t$ of all RNNs were 8-dimensional and were initialized based on the first input using a 2-layered MLP $f_{\text{init}}$ (neurons per layer: $16 \rightarrow 8$). All RNNs used a 3-layered preprocessing $f_{\text{pre}}$ (neurons per layer: $32 \rightarrow 16 \rightarrow 8$) and a linear mapping $f_{\text{post}}$ from the RNN cell output to the overall output.

During planning, the goal was to move the robot to the goal area (distance $< 0.15$) within 50 time steps. For model-based planning, we used iCEM [51]. We left the default hyperparameters as outlined in Pinneri et al. [51], but used the same planning horizon of 50 time steps as during training and simulated 256 trajectories per optimization step. Additionally, we used colored noise with $\beta = 3$. The cost was defined as the distance between robot and goal area. We found that iCEM, which was previously used with the ground truth simulator as a model [51], was relatively sensitive towards model errors, resulting in the agent often slightly missing the computer or stepping over it without activating the robot. To avoid floor effects based on the planning method, we simplified the task during planning by increasing the radius to interact with the computer by $50\%$.

### B.4  Shepherd

In the Shepherd scenario, illustrated in Fig. 8, an agent's goal is to catch a sheep using a portable cage. The agent's actions $a_t \in [-1, 1]^3$ control the agent's two-dimensional movement and whether the cage is grasped and carried ($a_t^3 > 0$) if it is in proximity. When the cage is carried, it moves with the agent. In every sequence, a sheep starts at the upper side of the scene (blue line in Fig. 8). The

---

[5]ODE, available at http://www.ode.org/, is licensed under the GNU Lesser General Public License version 2.1 as published by the Free Software Foundation.

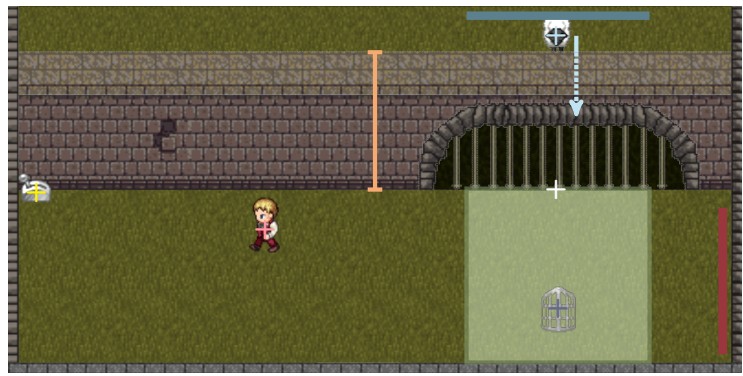

Figure 8: Shepherd scenario. Relevant positions are marked by a plus (+), with the lever in yellow, the agent in pink, the sheep in cyan, its reapearrance position in white, and the cage in purple. The orange bar visualized the wall height. The blue line and red line illustrate the sheep's and agent's starting position, respectively. The green area illustrates the cost function used for model-based planning. See text for more details.

sheep moves downwards with a randomly selected velocity, i.e. only changing its $y$-position (cyan arrow in Fig. 8). Thereby, the horizontal $x$-position of the sheep remains the same. Once the sheep reaches a wall, its position is occluded from the observation. The height of the wall (orange bar in Fig. 8) varies between simulations. The agent can make the sheep reappear again by activating a lever at a fixed position (yellow + in Fig. 8). The lever is activated once the distance of the agent to the lever is below a certain interaction threshold. As a result, a gate in the wall opens, causing the sheep to appear at the same horizontal position as before but at a lower vertical position (white + in Fig. 8). After its reappearance, the sheep moves downwards with the same velocity as before. It stops moving if it reaches the cage (distance below a certain threshold) or if it reaches the lower border of the scene. Observation $o_t \in [-1, 1]^7$ contains the agent's position (pink + in Fig. 8), the sheep's position (cyan +), the cage's position (purple +), and the height of the wall (orange bar). When the sheep is occluded, its position is masked by replacing it with a fixed value outside the normal range of coordinates.

We generated a dataset of 100 time step sequences by using randomly sampled actions. In 75% of the sequences up- and left-movements were sampled more frequently to get the agent to activate the lever. The dataset was split into training data (12.8k sequences), testing data (12.8k sequences), and validation data (6.4k sequences). To balance the datasets across possible events, we ensured that in each dataset during 75% of the sequences the lever was activated and in 25% of the sequences the sheep was caught in the cage. We trained the networks using minibatches of size 128 for 10k epochs. We used scheduled sampling [49] as a training regime and exponentially decreased the sampling probability $p_i$ to a minimum value of $p_{\min} = 0.05$.

All RNNs used 8-dimensional latent states $h_t$. The latent state $h_0$ was initialized based on the first two inputs using a 3-layered MLP $f_{\text{init}}$ (neurons per layer: $64 \rightarrow 32 \rightarrow 16$). All RNNs used a 3-layered preprocessing $f_{\text{pre}}$ (neurons per layer: $64 \rightarrow 32 \rightarrow 16$) and a linear mapping $f_{\text{post}}$ as a readout layer.

During planning, the agent started on the right side of the environment (red line in Fig. 8) holding the cage. The agent had 60 time steps to place the cage, move to the lever to open the gate, and let the sheep enter the previously placed cage. We chose a very short time of 60 time steps for this task to eliminate time-consuming solutions that avoid predicting the occluded sheep's future position, e.g. by catching the slowly moving sheep after its reappearance by going back and replacing the cage. For model-based planning, we used iCEM [51] with the same parameters as in Suppl. B.3 but predicting for a longer planning horizon of 100 time steps as during training. The cost was defined as the distance between the sheep and the cage, which was clipped to a large constant value when the sheep was above the gate (i.e. outside of the green area in Fig. 8). As in Suppl. B.3, we increased the interaction radius of the lever and the cage during planning by 50%.

## B.5 Fetch Pick&Place

Fetch Pick&Place is a benchmark reinforcement learning environment of OpenAI Gym[6] [47]. In Fetch Pick&Place a 7 DoF robotic arm with a two-fingered gripper is position-controlled through its four-dimensional action. The state of the scenario $s_t \in \mathbb{R}^{25}$ is composed of the positions of the endeffector and the object, the relative position between endeffector and object, the distance of the fingers to the center of the gripper, the rotation of the object, and the positional and rotational velocities of the endeffector, the object, and the fingers. To make the scenario partially observable, we omitted positional and rotational velocities as well as the rotation of the object in the observation $o_t \in \mathbb{R}^{11}$. The four-dimensional actions $a_t \in [-1, 1]^4$ control the three-dimensional position of the endeffector and the closing or opening of the fingers. Internally, the position control of the endeffector is realized by a PID-controller that runs at a higher frequency.

We generated our data consisting of sequences using APEX [57], a policy-guided model predictive control method, which was trained to move the object to a random goal position. APEX was deployed using the ground truth simulator as the internal model and hyperparameters as detailed in Pinneri et al. [57].

APEX finds various, surprisingly creative ways to move the object to the goal position, including pushing, sliding or flicking the object. For the experiments on policy generalization (Sec. D.7), we only considered sequences in which the object was grasped and lifted. Thus, we excluded all sequences in which the object moved while not being inside the gripper. For training and testing we considered 3.84k sequences wit a length of 25 time steps, in which the hand graps the object after at $t = 5$. A grasp was only considered if the relative $x-$ and $y-$ distance to the gripper was less than $0.0005$ and the relative $z-$distance was below $0.15$. Additionally, the object must not have changed its position before $t = 5$ to exclude sequence in which the object was pushed before. We randomly split this dataset into a training (3.2k) and testing set (640). For the generalization set we used 3.2k randomly selected sequences in which the grasp occurs at a later time $t$ with $t \in [6, 10]$.

In an additional experiment outlined in Suppl. D.8 we train the networks on all kinds of sequences. For that, we randomly split the collected dataset without further filtering into training (12.8k sequences), validation (6.4k sequences) and testing (6.4k sequences) sets. Here we considered sequences that are 50 time steps long.

In both experiments we trained the networks using minibatches of size 128 for 5k epochs using scheduled sampling [49], where we exponentially decreased the sampling probability $p_i$ to a minimum value of $p_{\min} = 0.05$. The latent state $h_t$ of all RNNs was 16-dimensional. The first latent state $h_0$ was initialized based on the initial input $(o_1, a_1)$ using a 2-layered MLP $f_{\text{init}}$ (neurons per layer: $32 \rightarrow 16$). All RNNs used a 3-layered preprocessing $f_{\text{pre}}$ (neurons per layer: $64 \rightarrow 32 \rightarrow 16$) and a linear mapping two-layered MLP $f_{\text{post}}$ to the network output. In the experiment using simpler, filtered data (Suppl. D.7) we used one-layered RNN cells. For the diverse data set (Suppl. D.8) we use stacked RNN cells (3 layers) and GateL0RD with 3-layered $g-$ and $r-$functions.

## B.6 Reinforcement Learning: General Training Principles and Hyperparameter Search

For our reinforcement learning experiments in the Mini-Gridworld [48], we used an actor-critic architecture as previously done by Chevalier-Boisvert et al. [52].[7] The architecture is a modified version of our general architecture (Fig. 1d), shown in Fig. 9. The image-like input is preprocessed by a three-layered convolutional neural network $f_{\text{pre}}$ with $2 \times 2$ convolution kernels and with max-pooling after the first layer. The 64-dimensional image embdeding is processed by an LSTM with 64-dimensional latent state. The LSTM output is processed by two separate MLPs, akin to using two $f_{\text{post}}$ in Fig. 1d, that take the role of the actor and the critic. The actor MLP $f_{\text{actor}}$ outputs the policy $\pi_t$, which determines the next action $a_t$. The critic MLP $f_{\text{critic}}$ outputs a value estimate $v_t$. Both MLPs use two layers with 64 neurons on the intermediate layer. In our experiments with GateL0RD, we only replace the LSTM cell and leave $f_{\text{pre}}$, $f_{\text{actor}}$, and $f_{\text{critic}}$ unmodified.

As done by Chevalier-Boisvert et al. [52], we train the system using Proximal Policy Optimization (PPO) [53] with parallel data processing. We performed 4 epochs of PPO with a batch size of 256. We

---

[6]OpenAI Gym is released under MIT license.

[7]We used an implementation by one of the authors available at `https://github.com/lcswillems/rl-starter-files`. The code is licensed under MIT license.

took the PPO hyperparameters from [52], setting $\gamma = 0.99$ and the generalized advantage estimation to 0.99.

We train the system using Adam [46] with $\beta_1 = 0.9$, $\beta_2 = 0.999$, and $\epsilon = 0.0001$. To determine the learning rate $\alpha$ we ran a grid search on the vanilla system (LSTM) with $\alpha \in \{0.005, 0.001, 0.0005, 0.0001\}$ for two random seeds and compared the mean rewards after training. In five of the six environments $\alpha = 0.001$ achieved the best results. Thus, for consistency we ran the MiniGrid experiments with a learning rate of $\alpha = 0.001$. For the one environment (KeyCorridorS3R2) in which a smaller learning rate ($\alpha = 0.0005$) produced better results, we additionally evaluated the system with the optimized learning rate and report the results in Suppl. D.9. As before, we apply gradient norm clipping [56] with a clipping threshold of 0.1. The loss was backpropagated for 32 time steps.

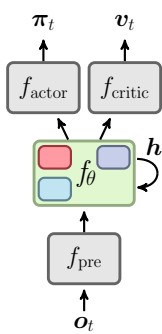

Figure 9: Reinforcement learning architecture

When using GateL0RD, we simply replaced the LSTM cell and left all hyperparameters unmodified. The PPO loss [53] was used as $\mathcal{L}_{\text{task}}$ in Eq. 2 and we set $\lambda = 0.01$ in all experiments. All reinforcement learning experiments were run with 10 random seeds per configuration.

### B.7 MiniGrid

MiniGrid [48] is a library of partially-observable benchmark reinforcement learning problems.[8] All MiniGrid environments consist of a $N \times M$ tiles. Each tile can be empty or contain one entity such as keys, doors, or walls. The agent receives an image-like, egocentric view of the $7 \times 7$ tiles in front of the agent. For each tile the agent receives a 3-dimensional signal, describing what type of object is in this tile, the color of the object, and its state (e.g. open, closed, or locked doors). The agent can't see through walls or closed doors. In every time step the agent can perform one of the following actions: move forward, turn left, turn right, pick-up an object, drop-off an object, or interact with an object (e.g. open doors). In all environments a sparse reward of 1 is given once the task is fulfilled. In some environments the time to fulfill a task is used to discount the rewards. Figure 10 shows all the problems we consider.

In **DoorKey-8x8** (Fig. 10a) the agent needs to move to the green square behind a locked yellow door. The agent needs to learn to pick up a yellow key to open the door. The environment is $8 \times 8$ tiles big but the size of the two rooms varies per simulation. **DoorKey-16x16** (Fig. 10g) is the same problem but in a larger $16 \times 16$ environment. We use the larger version to test zero-shot generalization, by training the system on the smaller environment and testing it on the larger one (see Sec. D.10).

In **RedBlueDoors-8x8** (Fig. 10b) the agent is randomly placed in a room ($8 \times 8$ tiles) with a red and a blue door. The agent has to first open the red door and afterwards open the blue door. Opening the blue door first result in ending the simulation without any reward.

In **SimpleCrossingS9N3** (Fig. 10c) the agent needs to navigate through a maze to a green square in the bottom left corner. The maze is randomly constructed by three walls that run horizontally or vertically through the room. Each wall has a single gap. **LavaCrossingS9N2** (Fig. 10d) poses the same problem, however, the walls of the maze are replaced by two lava rivers. Lava rivers do not occlude the view but entering lava terminates the episode without rewards. Because of the early terminations and sparse rewards, this environment is much more challenging to learn than the maze with walls.

In **KeyCorridorS3R2** (Fig. 10e) the agent needs to pick up a ball. The ball is locked behind a door and the key is hidden in some other room. Thus, the agent needs to learn to explore the rooms, by opening differently colored doors, to find the key. The agent can only pick up the ball if the agent is not holding the key, so after unlocking the door leading to the ball, the agent needs to drop the key.

**MemoryS13Random** (Fig. 10f) is a memory task. Here the agent needs to memorize a green object (key or ball) in one room, move through a corridor, and then either go left or right to the matching object. The environment is $13 \times 13$ tiles big. The length of the corridor is randomly generated per

---

[8]MiniGrid is available at `https://github.com/maximecb/gym-minigrid`. MiniGrid is licensed under Apache License 2.0.

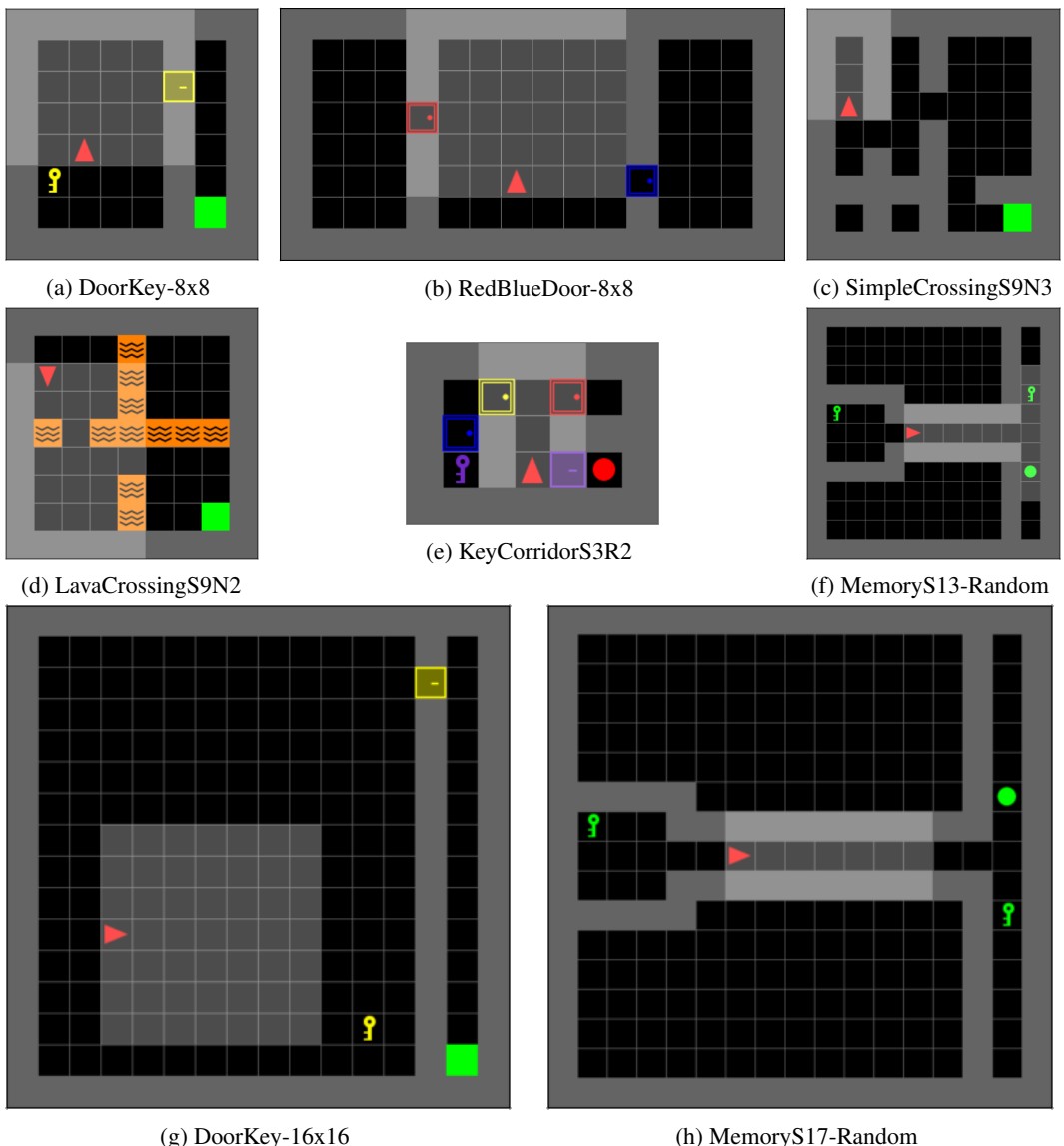

(a) DoorKey-8x8     (b) RedBlueDoor-8x8     (c) SimpleCrossingS9N3

(d) LavaCrossingS9N2

(e) KeyCorridorS3R2

(f) MemoryS13-Random

(g) DoorKey-16x16     (h) MemoryS17-Random

Figure 10: All MiniGrid envrionments used in this work.

run. In **MemoryS17Random** (Fig. 10h) the same problem needs to solved, but the environment is bigger ( $17 \times 17$ tiles). We use this version to test zero-shot generalization, by training the system on the smaller environment and testing it on the larger one (see Sec. D.10).

## B.8 Code and Computation

The code to run our experiments can be found at `https://github.com/martius-lab/GateL0RD` All experiments were run on an internal CPU cluster. Robot Remote Control experiment using GateL0RD take between 3-4 hours run time. Billiard Ball and Fetch Pick&Place experiments, which use larger datasets, take around 6-9 hours run time for GateL0RD. Shepherd simulations, which we train for twice the number of epochs, take approximately 18-22 hours of run time. The MiniGrid experiments vary largely in their training time and took between 2 and 30 hours to train. The baseline RNNs are roughly a factor of 0.8 faster than GateL0RD. This is mainly due to their optimized implementation in PyTorch.

# C  Ablation studies

In this section, we investigate the importance of each of the components of our proposed architecture.

## C.1    Ablation 1: Ablation of the type of gate function

We use the Billiard Ball scenario, trained using scheduled sampling [49] as in Sec. 6.1, to analyze the effect of different gate activation functions. In one ablated setting, we replace our ReTanh activation $\Lambda$ in Eq. 8 with a sigmoid activation function $\sigma$. Additionally, we test using the Heaviside step function $\Theta$ as gate activation function in Eq. 8. When using the Heaviside step function, we estimate the gradients using the straight-through estimator [27], which treats the step function as a linear function during the backward pass (illustrated in Fig. 1c). We test the Heaviside gates both with our $L_0$ loss ($\lambda = 0.001$) and without latent state regularization ($\lambda = 0$). Because a gate output of 0 is practically not achieved for the sigmoid function, we test the sigmoidal gates without latent state regularization ($\lambda = 0$).

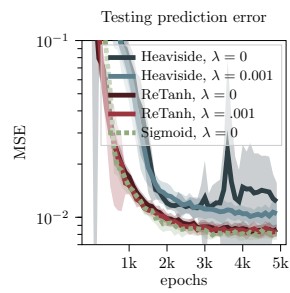

Figure 11: Billiard Ball testing error for GateL0RD with different gate functions.

Figure 11 shows the autoregressive prediction errors of the ablated versions of GateL0RD. The ablations with Heaviside gates perform worse than GateL0RD with the non-binary gates. When using the Heaviside gate without any regularization, the mean prediction error even increases over training time. GateL0RD with a sigmoid gate and our ReTanh gate reach the same level of prediction accuracy.

We believe that the worse performance of the Heaviside gate is due to the network profiting from multiplicative computations when computing the next latent state. For the Heaviside gate, interpolations of old and new latent states are not possible. Here, the latent state is either completely replaced or left unmodified. We conclude that our novel ReTanh gate is as suitable for gating as the classically used sigmoid gate. Additionally, it has the practical advantage of achieving an output of exactly 0, thus allowing the gate activation to be regularized as we do it with our $L_0$ loss.

## C.2    Ablation 2: Effect of gate stochasticity

To ablate the effect of the gate noise we compare GateL0RD with different strengths of the gate noise. For deterministic gates we set $\epsilon = 0$ in Eq. 10. Additionally we compare two values for the noise variance $\sigma$ of the diagonal covariance matrix $\Sigma$ in Eq. 10. We test the effects of gate stochasticity for a fixed value of gate regularization $\lambda = 0.01$ in the Billiard Ball task.

Figure 12a shows the prediction errors comparing deterministic gates to stochastic gates with different gate noise. There is no noticeable difference in prediction accuracy between the different settings. Thus, reasonable values of noise on the gate input during training does not noticeably affect the prediction error during testing.

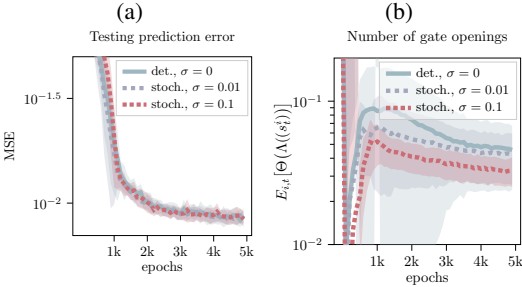

Figure 12: Billiard Ball task comparing effect of gate noise on GateL0RD ($\lambda = 0.01$): prediction error (a) and mean number of ate openings(b). Shaded areas denote standard deviation.

Figure 12b shows the average latent state changes per sequence, computed as $\mathbb{E}_{i,t}\left[\Theta\left(\Lambda(s_t^i)\right)\right]$, for all settings. Here, a larger value of gate noise results in fewer gate openings and thus, in fewer changes in the latent state.

We conclude that using stochastic gates together with our $L_0$ loss has a regularizing effect: GateL0RD trained with stochastic gates seems to achieve the same level of prediction accuracy as when trained with deterministic gates but changes its latent states more sparsely.

## C.3 Ablation 3: Ablation of the latent state initialization network $f_{\text{init}}$

Next we ablate the effect of the context network $f_{\text{init}}$, which sets the latent state based on a few initial inputs (see Fig. 1d). We compare all RNNs against variants without $f_{\text{init}}$ in the Billiard Ball scenario. When omitting $f_{\text{init}}$, we initialize the latent state with $\boldsymbol{h}_0 = \boldsymbol{0}$.

Figure 13 shows the prediction errors for all RNNs when using the context network $f_{\text{init}}$ (solid lines) and when initializing the latent state with zeros (dotted lines). The prediction accuracy decreases for all network types when trained without the

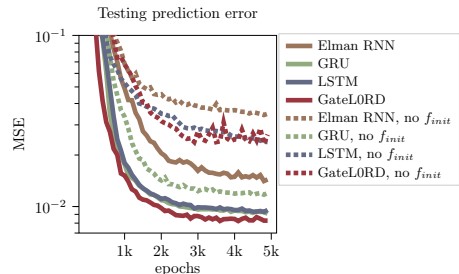

Figure 13: Billiard Ball: effect of latent state initialization with or without $f_{\text{init}}$ on prediction errors.

context network. However, how much their performance drops varies across the different RNN types. GRUs seem to be much less affected by using them without $f_{\text{init}}$ than LSTMs and GateL0RD ($\lambda = 0.001$).

## C.4 Ablation 4: Ablation of the output function

After updating its latent state $\boldsymbol{h}_t$, GateL0RD uses two one-layered MLPs $p$ and $o$ to compute the network output as $p(\boldsymbol{x}_t, \boldsymbol{h}_t) \odot o(\boldsymbol{x}_t, \boldsymbol{h}_t)$ (see Eq. 9). With this output function we want to enable both additive as well as multiplicative effects of the latent state $\boldsymbol{h}_t$ and input $\boldsymbol{x}_t$ on the network output. Is this justified or would a simple MLP as output function suffice?

We analyze the effect of our output function in the Robot Remote Control Scenario ($\lambda = 0.001$, trained on random action rollouts $\mathcal{D}_{\text{rand}}$). Here we compare GateL0RD using our standard output function ($p(\boldsymbol{x}_t, \boldsymbol{h}_t) \odot o(\boldsymbol{x}_t, \boldsymbol{h}_t)$) to an ablated version using just a one-layered MLP with $\tanh$ activation ($p(\boldsymbol{x}_t, \boldsymbol{h}_t)$).

Figure 14 shows the resulting prediction errors of GateL0RD using its normal output function compared to the case without a multiplicative gate ($p(\boldsymbol{x}_t, \boldsymbol{h}_t)$). Clearly GateL0RD achieves a much better prediction when using a multiplicative output gate instead of a sim-

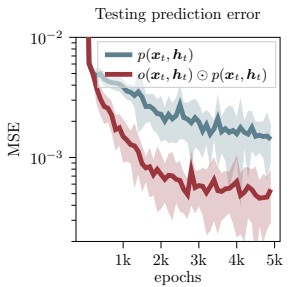

Figure 14: Robot Remote Control: GateL0RD with an output function containing a multiplication ($p \odot o$) or not ($p$)

ple MLP. Thus, a multiplicative branch for computing the network output seems to improve the prediction accuracy. This may also explain the worse prediction accuracy of Elman RNNs in most tasks since they lack the multiplicative gates that can be found in all other investigated RNNs.

## C.5 Ablation 5: Comparison against $L_1/L_2$-versions

Our hypothesis is that sparsely changing latent states allows better generalization across spurious temporal dependencies in the training data. GateL0RD enforces such a sparsity of latent updates via an $L_0$-regularization of the changes in latent state. This is implemented using the novel ReTanh gate, instead of the commonly used sigmoid gates, and an auxiliary $L_0$ loss term that is made differentiable using the straight through estimator. Is this necessary or would a simple sigmoid gate in conjuction with an $L_1$ or $L_2$ loss also improve generalization?

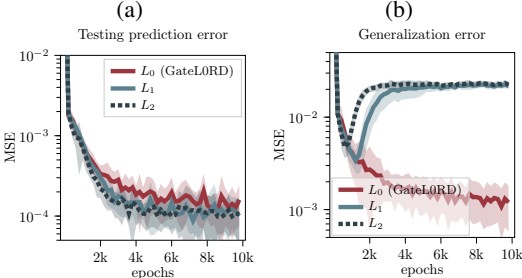

Figure 15: Robot Remote Control: prediction error on the test set (a) and on the generalization set (b) for GateL0RD, $L_1$-, and $L_2$-variants.

To analyze this, we compare GateL0RD against ablated versions that use a sigmoid gate and penalize the $L_1$ or $L_2$ norm of the gate activations. We compare the version in the Robot Remote

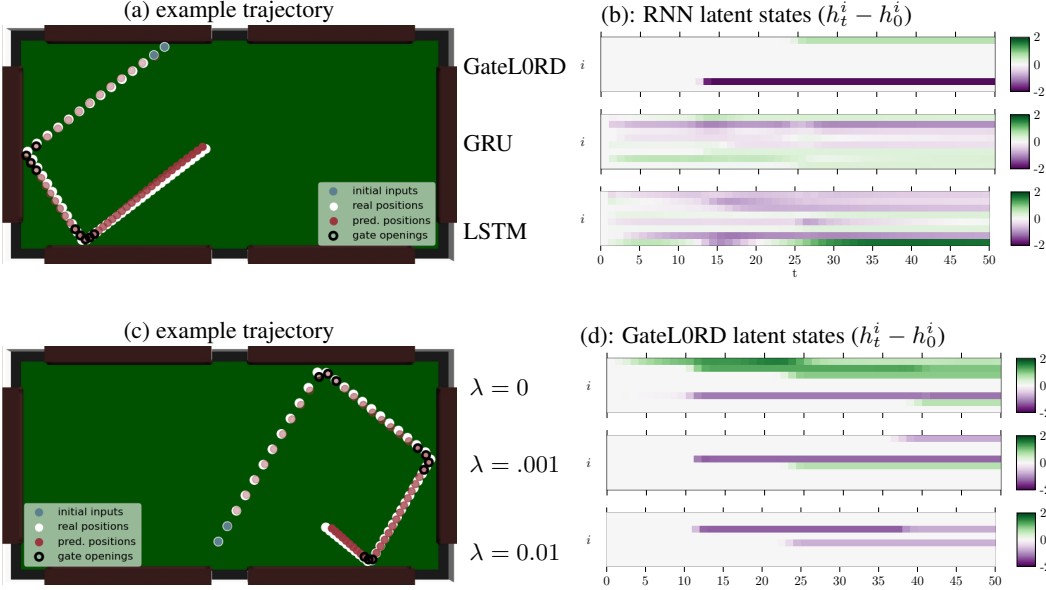

Figure 16: Billiard Ball example trajectory. (a) & (c): Real positions are shown in white, the provided inputs in blue, GateL0RD ($\lambda = 0.01$) position predictions in red (saturation increasing with time). The inputs for which at least one gate opened are outlined in black. (b): The latent states $\boldsymbol{h}_t$ for different RNNs for the sequence shown in (a). (d): The latent states $\boldsymbol{h}_t$ for GateL0RD with different values of $\lambda$ for the sequence shown in (c). Latent states are shown relative to the initial latent state $\boldsymbol{h}_0$.

Control setting as in Sec. 6.2. Thus, we train the networks on random action rollouts with linearly increasing action magnitude and test it either on data generated by the same process (testing) or on uniformly sampled random actions (generalization). We chose a suitable regularization hyperparameter $\lambda = 0.001$ for all variants.

Figure 15a shows the prediction errors during testing for all variants. The $L_1$- and $L_2$-ablations achieve a very low prediction error on the test set, even exceeding GateL0RD's prediction in terms of accuracy. However, when tested on the generalization set, shown in Fig. 4b, their prediction error increases drastically.

We conclude that the $L_1/L_2$-variants behave similar to GRUs and LSTMs (compare Fig. 4b and Fig. 15b). They achieve a low testing error but fail to generalize to data generated by a different policy. This suggests that they also strongly overfit to spurious temporal dependencies, unlike our $L_0$-version.

However, it is noteworthy that on the test set the $L_2$-variant manages to achieve the lowest mean prediction error of all investigated RNNs. Krueger and Memisevic [35] previously suggested to penalize the $L_2$ norm of latent state changes in RNNs to prevent exploding or vanishing activations. Our results suggest that applying $L_2$-regularization on the latent state changes seems to be a promising approach to increase the in-distribution performance of RNNs.

# D  Additional experiments and analysis

## D.1  Billiard Ball: Analyzing the latent states and gate usage

In this section, we provide further exemplary latent states for RNNs when applied to the Billiard Ball scenario. Figure 16 shows two exemplary ball trajectory and the corresponding latent states. GateL0RD is able to make accurate autoregressive predictions (see red dots in Fig. 16a and Fig. 16c) and tends to open its gates around wall collisions (black circles). Figure 16b shows the latent states of GateL0RD ($\lambda = 0.01$) compared to the latent states of a GRU and a LSTM for the trajectory shown in Fig. 16a. GateL0RD's changes in latent states are easily interpretable: GateL0RD seems

to encode $x-$ and $y-$ velocity in two dimensions of its latent state and changes the latent state at these particular dimensions when the ball velocity changes upon collision. The LSTM and GRU also tend to change their latent states more around points of collision but also change many latent state dimensions throughout the trajectory, making them much harder to interpret.

Figure 16d shows the latent states of GateL0RD for the same sequence, shown in Fig. 16c, using different values of the sparsity regularization hyperparameter $\lambda$. As before, GateL0RD with $\lambda = 0.01$ uses two dimensions of its latent state to encode the ball velocity and updates these two dimensions upon collisions. In this example, GateL0RD with $\lambda = 0.001$ uses three dimensions to encode the ball's velocity. With every collision a different latent state dimension is updated, instead of using the same dimension for changes in $y-$velocity, as done by GateL0RD with $\lambda = 0.01$. In this example, GateL0RD with $\lambda = 0$ uses five dimensions to encode $x-$ and $y-$ velocities. At points of collision, multiple latent dimensions change.

To further illustrate how the regularization hyperparameter $\lambda$ affects the latent state changes, we plot the number of latent state dimensions that change on average while predicting a Billiard Ball sequence in Fig. 17. As expected, a stronger regularization through $\lambda$ results in fewer dimensions of the latent state changing. Without regularization ($\lambda = 0$) GateL0RD changes on average less than 6 dimensions of the 8-dimensional latent state. For $\lambda = 0.01$, GateL0RD quickly converges to on average using two latent states. For $\lambda = 0.1$, fewer latent state dimensions change on average.

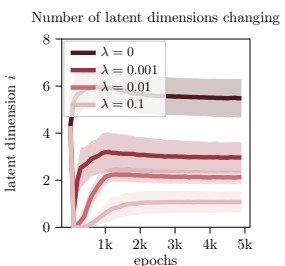

Figure 17: Billiard Ball: latent state dimension usage for different values of $\lambda$.

As shown in Fig. 3d, even without regularization ($\lambda = 0$) GateL0RD continuously decreases the mean number of gate openings. After 5k epochs, GateL0RD on average opens a gate less than $50\%$ of the time. Similarly, it does not use all dimensions of its latent state, as shown in Fig. 17. This effect emerges from the interplay of stochastic gradient descent and the ReTanh having gradients of 0 for inputs $s_t^i \leq 0$. Over training time, gates will randomly close and kept shut if they do not contribute to decreasing the loss. This effect is closely related to the "dying ReLU problem" when using ReLU activation functions [58]. While dying ReLUs are considered a problem, in our case this is advantageous whenever the gate regularization is beneficial. We believe that this results in GateL0RD, even without regularization, being more robust to out-of-distribution shifts than GRUs and LSTMs. For example, GateL0RD with $\lambda = 0$ achieves a smaller mean autoregressive prediction error when trained using teacher forcing (Fig. 3a), compared to the baseline RNNs.

### D.2 Robot Remote Control & Shepherd: Loss and scheduled sampling

In Fig. 18b we provide the loss curves for the Robot Remote Control scenario and Fig. 18c shows the loss curves for the Shepherd task. For both tasks the loss decreases during the first couple of epochs, increases again until roughly 2k epochs, and continuously decreases afterwards. This development is caused by using scheduled sampling [49] as a training regime (detailed in Suppl. B.1). The probability $p_i$ of applying teacher forcing exponentially decreases over the first 2k epochs, as shown in Fig. 18a for the Robot Remote Control task. Thus, over the first 2k epochs the problems change from 1-step

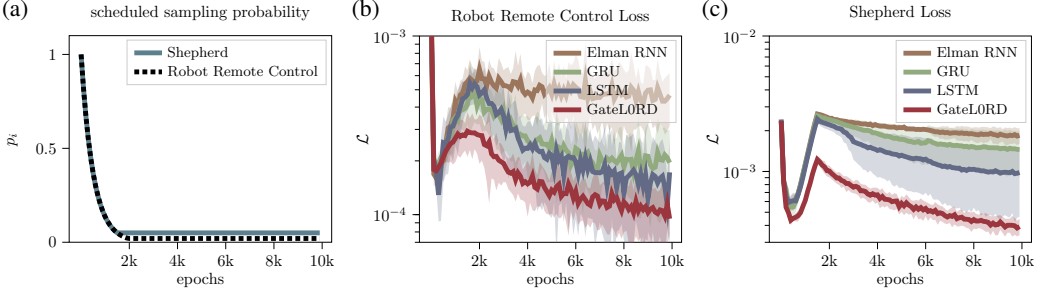

Figure 18: Scheduled sampling and loss curves: Probability $p_i$ of using the real input instead of the predicted input (a). Loss curves in Robot Remote Control (b) and Shepherd (c). Shaded areas show standard deviation.

prediction problems to $N$-step prediction problems. This drastically increases the difficulty within the first 2k epochs. However, this transition helps to learn autoregressive predictions [49] as also demonstrated by our Billiard Ball experiments (Sec. 6.1).

### D.3 Robot Remote Control: Improving RNN generalization

In Sec. 6.2 we showed that LSTMs and GRUs trained for the Robot Remote Control environment using data in which action magnitude was positively correlated with time ($\mathcal{D}_{\text{time}}$), failed to properly generalize to testing data without this correlation ($\mathcal{D}_{\text{rand}}$). GateL0RD showed less performance degeneration when tested on the generalization dataset. We hypothesized, that GateL0RD's superior generalization performance was based on its tendency to only encode unobservable information within the latent states, making it less prone to overfit to observable spurious temporal dependencies within the training data. However, an alternative explanation would be that the overfitting of LSTMs and GRUs was caused by their learning rate. To investigate if the other RNNs' generalization abilities can be improved to the level of GateL0RD by choosing a different learning rate, we ran a grid search over three learning rate values ($\{0.005, 0.001, 0.0005\}$) for LSTMs, GRUs, Elman RNNs with two

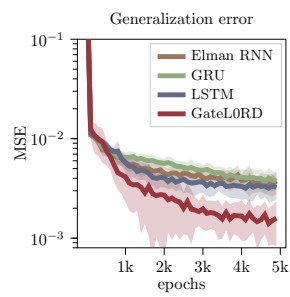

Figure 19: Robot remote control generalization prediction error with optimized learning rates.

random initializations. We selected the learning rate that lead to the lowest mean squared prediction error for the 50-timestep predictions on the validation dataset of $\mathcal{D}_{\text{rand}}$ after 5k epochs. Seeing that a learning rate of 0.001 yielded the best validation error for all RNNs, we reran the experiment with this learning rate (10 random seeds).

Figure 19 shows the resulting prediction error when testing the RNNs on the generalization test set of $\mathcal{D}_{\text{rand}}$. While the prediction error of GRUs and LSTMs on the generalization test set improved compared to our previous experiment, GateL0RD still achieved a lower prediction error on the generalization data than the other RNNs. Note that GateL0RD was not further optimized in this experiment. Thus, we conclude that GateL0RD's superior generalization performance in this setting is not caused by the learning rate.

### D.4 Robot Remote Control: Training on uniformly sampled random action rollouts

We previously showed for the Robot Remote Control environment that GateL0RD generalized better than the other RNNs to the data generated from random action rollouts ($\mathcal{D}_{\text{rand}}$) when trained on a dataset that contained spurious temporal correlations ($\mathcal{D}_{\text{time}}$) even for different learning rates. Besides GateL0RD better capabilities in generalization, another explanation could be that GateL0RD is simply better at predicting sequences from the particular dataset $\mathcal{D}_{\text{rand}}$. To rule out this alternative explanation this, we trained the RNNs on a training set, generated from uniformly sampled random action rollouts ($\mathcal{D}_{\text{rand}}$), and tested the network on data generated by the same process.

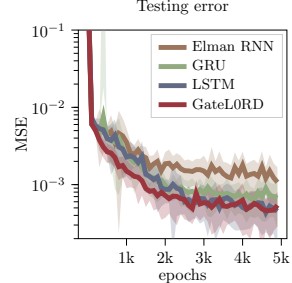

Figure 20: Robot remote control prediction error when trained and tested on $\mathcal{D}_{\text{rand}}$.

Figure 20 shows the testing prediction error for predicting sequences based on the first observation and a sequence of actions. After 5k epochs of training, LSTMs and GRUs achieve a similar prediction accuracy as GateL0RD ($\lambda = 0.001$). Thus, GateL0RD superior prediction accuracy on $\mathcal{D}_{\text{rand}}$ in previous experiments can indeed be attributed to its better generalization capabilities.

### D.5 Robot Remote Control: Learned latent states

In this section, we provide further exemplary latent states of the RNNs trained in the Robot Remote Control scenario as described in Sec. 6.2. Figure 21a shows one exemplary sequence in which the robot was controlled by the agent and the corresponding latent states for two instantiations of GateL0RD, GRU, and LSTM with different random seeds. GateL0RD seems to use one dimension of its latent state to encode when the agent controls the robot with its actions. For GRUs and LSTMs

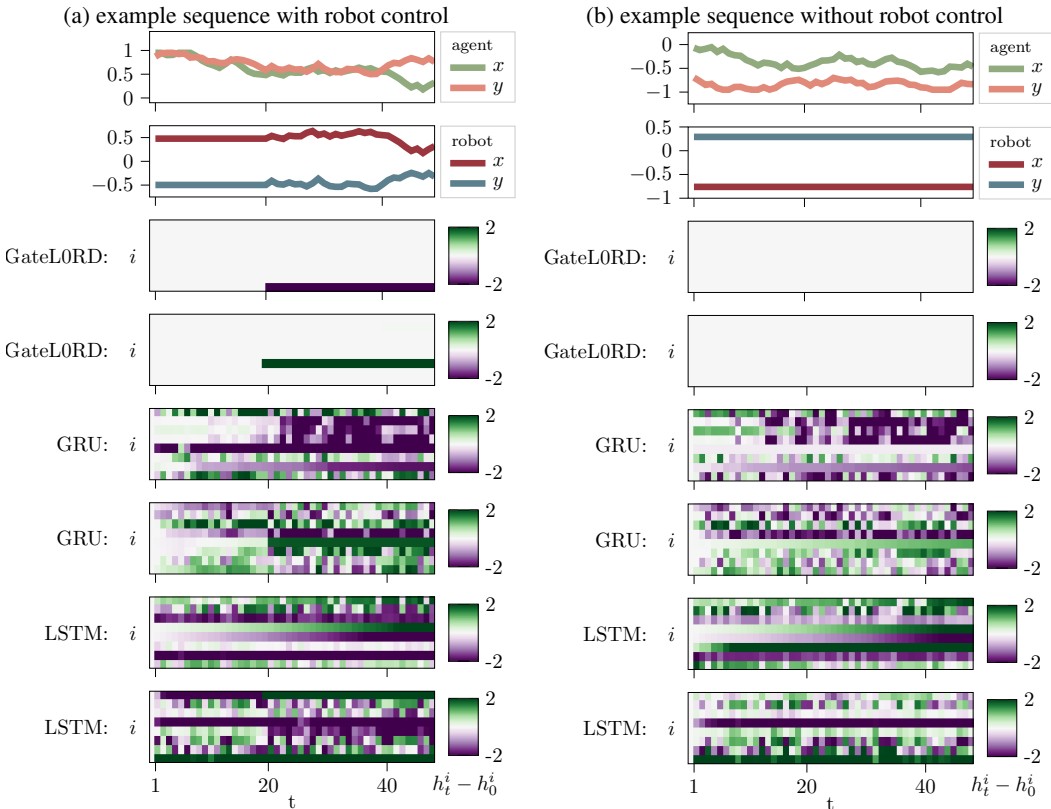

Figure 21: Latent state for two exemplary Robot Remote Control sequences in which the robot was either controlled (a) or not (b). The latent states $h_t$ are shown relative to their initialization $h_0$. We provide the latent states for two GateL0RDs, GRUs, and LSTMs (different random seeds). One row shows the same random seed.

the latent states also seem to strongly change around the point where the agent gains control over the robot, however, their latent states are not as clearly interpretable. Figure 21b shows one exemplary sequence, in which the robot was not controlled. Here, GateL0RD does not modify its latent states, whereas LSTMs and GRUs continuously change their latent states over the course of the sequence.

Note that when the robot is not controlled, as in Fig. 21b, Robot Remote Control is fully observable. Thus, it seems that GateL0RD able to learn to distinguish observable from unobservable information and attempts to only update its latent state when the unobservable information changes. To quantitavely evaluate this claim, we feed in all generalization sequences and classify the gate usage and unobservable events of task. The inputs of the sequences were classified based on whether control of the robot was triggered at this time step (control) or not (no control). Additionally we analyzed for each input whether one of GateL0RD's gates opened (gate open) or not (gate closed). The mean gate openings for the two events are shown in Table 2 with $\pm$ denoting standard deviation. GateL0RD seems to mostly open its gates when the robot is controlled and tends to keeps its gate shut at other time steps. Thus, GateL0RD indeed seems to mostly update its latent state when the unobservable state of the environment changes.

Table 2: Gating in Robot Remote Control

|  | gate open | gate closed |
|---|---|---|
| **control** | $0.978 \pm 0.016$ (hits) | $0.022 \pm 0.016$ (misses) |
| **no control** | $0.089 \pm 0.037$ (false alarms) | $0.911 \pm 0.037$ (correct rejections) |

Table 3: Robot Remote Control: final prediction errors

|            | testing                | generalization        |
|------------|------------------------|-----------------------|
| **CRNN**       | $0.00056 \pm 0.00015$ | $0.0331 \pm 0.0128$ |
| **Elman RNN**  | $0.00064 \pm 0.00036$ | $0.0062 \pm 0.0040$ |
| **GRU**        | $0.00030 \pm 0.00036$ | $0.0118 \pm 0.0058$ |
| **LSTM**       | $0.00018 \pm 0.00013$ | $0.0173 \pm 0.0058$ |
| **GateL0RD**   | $0.00015 \pm 0.00010$ | $0.0011 \pm 0.0007$ |

### D.6  Robot Remote Control: Clockwork RNNs

In the tasks we considered, the latent states need to change at irregular times and are depending on the state of the environment. Thus, we hypothesize that RNNs operating on predefined time scales, such as Clockwork RNNs [33], are not well suited for these tasks. We evaluate this in the Robot Remote Control task using Clockwork RNNs (CRNN, 3 clock modules, clock rates $T_1 = 1, T_2 = 4, T_3 = 8$). The learning rate ($\alpha = 0.001$) was determined via a grid search with $\alpha \in \{0.005, 0.001, 0.0005, 0.0001, 0.00005\}$. Unlike the other RNNs, the CRNN did not fully converge after 10k epochs, thus, instead, we trained it for 20k epochs.

In Table 3 we list the mean prediction error after full training (20 random seeds, $\pm$ denotes standard deviation) on the test and generalization set, compared to the other RNNs. CRNNs behave similarly to the other RNN baselines in that they achieve a reasonable test prediction error. However, they overfit even more drastically to the temporal correlations of the actions in the training set, resulting in a high prediction error on the generalization set.

### D.7  Fetch Pick&Place: Generalization across grasp timings

In Sec. 6.2 we showed using the Robot Remote Control scenario that GateL0RD is better at generalizing across spurious temporal dependencies in the training data than other RNNs. In a follow-up experiment we want to investigate if similar effects can be found in a more complex environment, using more natural training data. For that we use the **Fetch Pick&Place** environment and train the networks to predict reach-grasp-and-lift sequences. The training sequences were generated by a policy-guided model-predictive control method [57]. Importantly, we train the network only on sequences where the gripper first touches the object exactly at time $t = 5$. We test the networks on predicting sequences where gripper-object contact occurs as during training (testing set) or on sequences where the object is grasped later (generalization set).

Figure 22a shows the mean prediction errors during testing. All networks achieve a very low prediction error. The prediction accuracy is similar for all RNNs, but LSTMs achieve a slightly lower prediction error than GateL0RD ($\lambda = 0.0001$). When the networks were tested on sequences with different grasp timings, they produce much higher prediction errors as shown in Fig. 22b. GateL0RD prediction accuracy does not drop as strongly as the accuracy of the other networks. Thus, as in the Robot Remote Control experiments, GateL0RD more robustly generalizes across spurious temporal correlations.

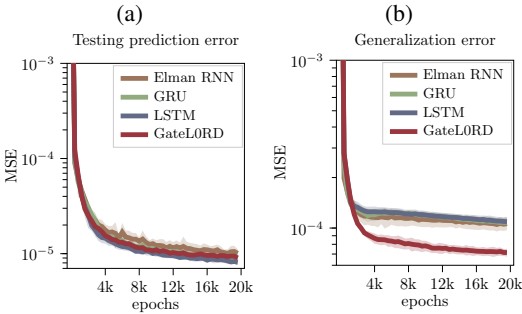

Figure 22: Fetch Pick&Place results: prediction error on test set (a) and generalization set (b). Shaded areas denote standard deviation.

Figure 23 shows the latent states of the different RNNs when predicting two exemplary sequences. Here, GateL0RD ($\lambda = 0.0001$) uses either one or three dimension of $\boldsymbol{h}_t$ that changes around the time when GateL0RD predicts that the gripper grasps the object. During the predicted transportation of an object, the latent state does not change anymore. This hints at GateL0RD encoding the event of "transporting an object" in one dimension of its latent state. For the other RNNs the latent state is not as easily interpretable.

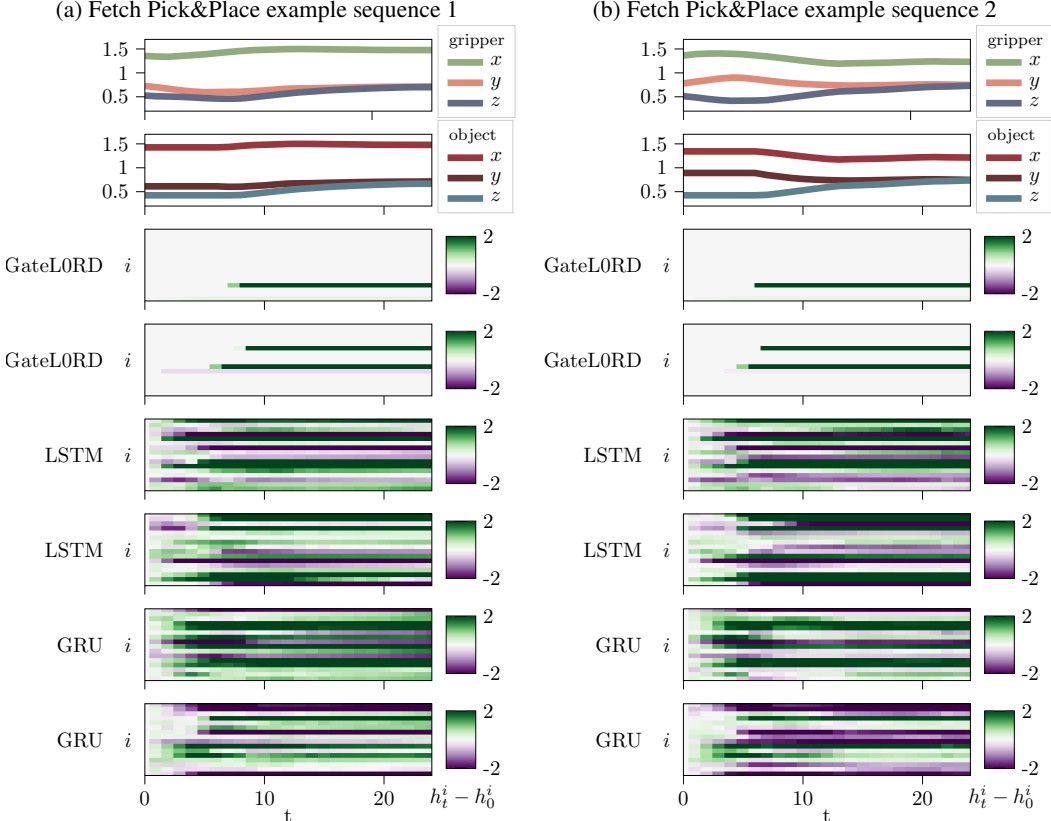

(a) Fetch Pick&Place example sequence 1    (b) Fetch Pick&Place example sequence 2

Figure 23: Latent states of different RNNs for two exemplary sequences in the Fetch Pick&Place environment. The latent states $h_t$ are shown relative to their initialization $h_0$. We compare the latent states for two random seeds each, where each row shows the same random seed.

## D.8    Fetch Pick&Place: Training on diverse sequences

Previously, we only considered reach-grasp-lift sequences in the Fetch Pick&Place environment. However, there are multiple other ways to move the object to a target position, such as pushing, sliding or even flicking. Thus, in a next experiment we analyze the performance of the RNNs when trained as a model on a diverse set of sequences generated by the policy-guided model-based control method APEX [57].

Figure 24 shows the prediction errors of the RNNs when predicting testing sequences given the first observations and sequence of actions. In this scenario, all RNNs achieve a very similar prediction accuracy. GateL0RD with $\lambda = 0.001$ produces a slightly higher mean prediction error than the other RNNs, whereas GateL0RD with $\lambda = 0$ achieves a slightly lower error. We believe that in this scenario the small differences in prediction accuracy are a result of better approximations of the endeffector velocities. In Fetch Pick&Place the position control of the endeffector is realized by a PID-controller running at a higher frequency, thus, in this scenario continuous latent state updates are advantageous for predicting the endeffector velocity. Hence, in this scenario $\lambda$ regulates the trade-off between prediction accuracy and latent state explainability and needs to be chosen depending on priorities of the application.

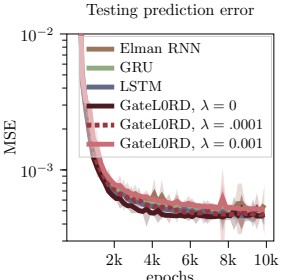

Figure 24: Fetch Pick&Place testing prediction errors.

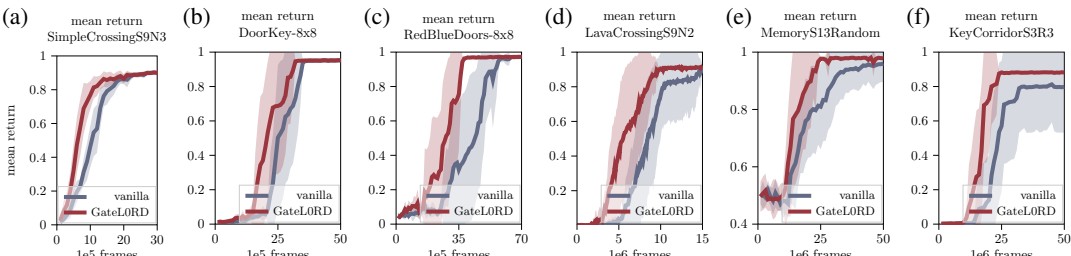

Figure 25: MiniGrid results: Mean rewards in solving various tasks when GateL0RD replaces an LSTM (vanilla) in a PPO architecture. Shaded areas show standard deviation.

## D.9 MiniGrid: Further analysis and experiments

In Sec. 6.4 we showed that GateL0RD is more sample efficient in achieving a high success rate in various MiniGrid tasks than when it replaces an LSTM in a PPO architecture. Some problems of MiniGrid discount the overall reward based on the number of actions required to reach the goal. Thus, another metric to judge success in MiniGrid is the mean reward collected by the systems. Figure 25 shows the mean rewards for the vanilla architecture and architecture containing GateL0RD over training experience. For all problems the architecture containing GateL0RD is more sample efficient and achieves high levels of reward faster.

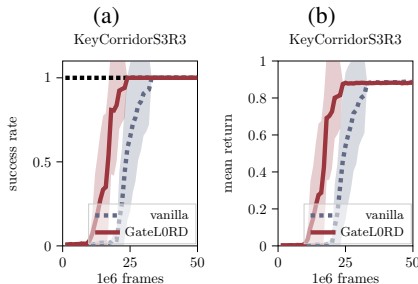

Figure 26: MiniGrid results: mean success rate and reward for KeyCorridorS3R3. The learning rate of the vanilla system was optimized for this problem.

For consistency we used the same hyperparameters in all MiniGrid experiments and only swapped the LSTM cell for GateL0RD. However, as described in Suppl. B.6 a grid search showed that for the KeyCorridorS3R3 problem a smaller learning rate ($\alpha = 0.0005$) resulted in higher mean rewards for the vanilla architecture. Thus, to exclude the possibility that GateL0RD outperformed the LSTM in this problem based on the choice of learning rate, we ran an additional experiment in the KeyCorridorS3R3 problem with the vanilla architecture using the optimized learning rate. The resulting mean success rate and mean rewards are shown in Fig. 26a and Fig. 26b, respectively. While the vanilla architecture now manages to reach a success rate of 100% and a mean reward larger than 0.8, GateL0RD is still faster in reaching the same level of performance.

## D.10 MiniGrid: Zero-shot policy transfer

We hypothesize that GateL0RD can memorize information precisely without information loss over time. Thus, it should be able to generalize well across different memory durations. We investigate this aspect in in the MiniGrid domain by training a PPO architecture containing an LSTM (vanilla) and the same architecture containing GateL0RD on two problems that require memory, i.e. DoorKey-8x8 (shown in Fig. 10a) and MemoryS13Random (shown in Fig. 10f). We evaluate the

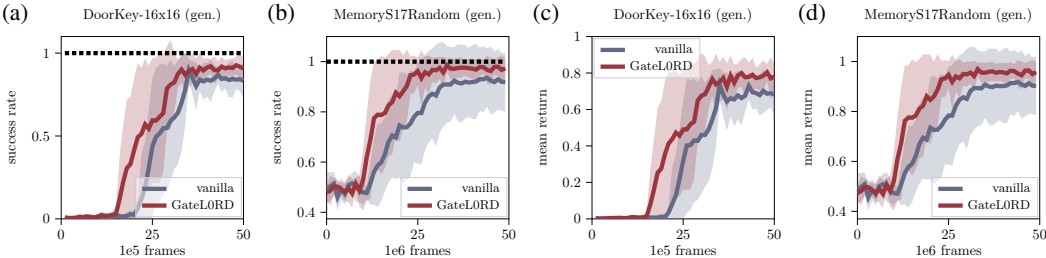

Figure 27: MiniGrid: zero-shot generalization for two environments when trained in a smaller versions of same problem. (a) & (b) show the success rate and (c) & (d) show the mean reward. Shaded areas show standard deviation.

architectures on the same problems in larger environments, i.e. DoorKey16x16 (shown in Fig. 10g) and MemoryS17Random (shown in Fig. 10h). Thus, one of the main challenges is that during transfer information needs to be memorized for longer periods of time.

Figure 27 shows the zero-shot generalization performance for solving the more complex problems after training only on the simpler variants. For both problems GateL0RD achieves a higher mean success rate and mean reward than the vanilla baseline. The better performance cannot simply be explained by GateL0RD being better at the considered task than the LSTM of the vanilla architecture. When tested in the simple problems both architectures achieve approximately the same performance (c.f., Fig. 25b). Instead the better performance is likely due to GateL0RD generalizing better from short-term to long-term memorization.