# OpenReview forum: "Sparsely Changing Latent States for Prediction and Planning in Partially Observable Domains"
_NeurIPS.cc/2021/Conference — NeurIPS 2021 Poster_

### Official Review · Reviewer_sDuQ · 2021-07-15

**Rating:** 7
**Confidence:** 4

**Summary:**

This paper presents a sequence prediction method that incorporates a particular inductive bias -- infrequency of latent state updates -- as measured by an L_0 term on latent state differences. This term is approximated and made more easily optimizable using a method similar to [16], which examined L_0 regularization in non-sequential domains. The paper conducts experiments on several low-dimensional POMDP tasks and then draws the following conclusions: the method generalizes better to out-of-distribution inputs in several partially observable domains than other commonly used RNNs, exhibits long term memory in a domain, and exhibits explainability (in its latent states), sometimes at the cost of reduced prediction performance.

**Limitations And Societal Impact:**

The limitations were appropriately described; the societal impacts were described inside the checklist (which are essentially a statement that there are no ethical concerns specific to the work). I agree with that sentiment, but it should be incorporated either into the main text or into the appendix, rather than the checklist, which will not be present in a potential final version(?)

**Main Review:**

**Opinion summary**: The paper offers a well-motivated method for sequence prediction in partially-observed domains that contain dynamics with significant changes. The proposed method is straightforward and clear. However, the method is of questionable originality (see below). The experiments are conducted relative to basic common NN-based sequence-prediction methods, which do show significant improvements, but lack comparison to previously-proposed methods for incorporating the same inductive bias (infrequency in latent state updating). Thus, we cannot conclude that this method is better than a relevant set of previous methods for accomplishing the same goals. There are a few other issues with the experiments that could be improved.

**The paper has questionable originality, as prior work accomplishes a similar goal in a different way**: there are missing comparisons to existing methods -- [25,26] use predefined time scales for gate opening, which may be sufficient for high performance on the considered tasks, and [27,28,29] all use binarized update gates, and are thus quite related to the proposed method. The proposed method is not clearly different enough from these methods, and is therefore of questionable novelty. I think empirical results that compare the methods are needed -- without them, we cannot understand situations in which the proposed method is better or worse than prior work that attempts to accomplish the same goal.

**The interpretability is not quantitatively supported:** The method pitches sparsity in gate opening as both a way to learn a more performant system, as well as a way to understand when the dynamics of the environment are significantly changing. On the latter point -- Figs 4d and 6b illustrate examples where the gate opening makes qualitative sense (the paper describes these as "interpretable"). These results provide intuition, but are not evidence of how reliable the method is at representing "significant and systematic" changes with gate openings. It would be interesting to have a better understanding of how reliable it is, for instance by (automatically?) labeling each point in a sequence as either a "significant and systematic" change or not (like the black-outlined positions in Fig 6a and the moving-robot positions in Fig 4d), and then reporting binary classification results (e.g. precision, recall, accuracy) using the classifier p(systematic change=1 | any gate open=1)=1; p(systematic change=0 | any gate open=0)=1.

**Other issues with experiments (ordered by decreasing priority):**
- L270 "only encodes unobservable information within its latent state" this claim has, at best, anecdotal evidence (Fig 4d). It is a significant claim to state generally, so either temper it or provide sufficient evidence to support it.
- The $\lambda=0$ results in Fig 3d requires further explanation. Why does the version with no regularization exhibit the fewest latent state changes (L248 says "a higher value results in fewer latent state changes.")?
- Fig 3c is too zoomed-out. The differences between the nonzero lambda experiments are difficult to see.
- Fig 15 is referenced in the text but appears in the supplement.

---
Post-response update: My above concerns have been addressed.


**Time Spent Reviewing:**

3

---

> ### Author Response · Authors · 2021-08-10
> **Answer to sDuQ**
>
> We thank the reviewer for their time and very helpful criticism.
>
> > there are missing comparisons to existing methods -- [25,26] use predefined time scales for gate opening, which may be sufficient for high performance on the considered tasks,
>
> In the tasks we considered, the latent states need to change at irregular times and are depending on the state of the environment. Thus, our assumption was that RNNs operating on predefined time scales, such as Clockwork RNNs [1], are not well suited for these tasks. But we agree that it is better to evaluate this empirically. We ran a new experiment in the Robot Remote Control (RRC) task (as in section 6.2) using Clockwork RNNs (CRNN, 3 clock modules, clock rates $T_1 = 1, T_2 = 4, T_3 = 8$). The learning rate (lr = 0.001) was determined via a grid search (as detailed in Suppl B.1). Unlike the other RNNs, the CRNN did not fully converge after 10k epochs, thus, instead, we trained it for 20k epochs. Below we report the mean prediction error (20 random seeds, +- denotes standard deviation) on the test and generalization set, compared to the other RNNs.
>
> ```
> Method    | Testing            | Generalization
> -------------------------------------------------
> CRNN      | 0.00056 +- 0.00015 | 0.0331 +- 0.0128
> Elman RNN | 0.00064 +- 0.00036 | 0.0062 +- 0.0040
> GRU       | 0.00030 +- 0.00036 | 0.0118 +- 0.0058
> LSTM      | 0.00018 +- 0.00013 | 0.0173 +- 0.0058
> GateL0RD  | 0.00015 +- 0.00010 | 0.0011 +- 0.0007
> ```
>
> CRNN behaves similarly to the other RNN baselines in that they achieve a reasonable test prediction error. However, they overfit even more drastically to the temporal correlations of the actions in the training set, resulting in a high prediction error on the generalization set.
>
> > [27,28,29] all use binarized update gates, and are thus quite related to the proposed method. [...] I think empirical results that compare the methods are needed
>
> We agree that the use of binary gates is highly related to our $L_0$ gates. We provided an ablation of GateL0RD using binary gates in Suppl C.1. As shown by Fig. 8, in the Billiard Ball scenario binary gates are not sufficient to achieve the same prediction accuracy as our ReTanh gate or sigmoidal gates. We believe more complex physical simulations require multiplicative gates and interpolations between latent states to achieve accurate predictions. This can be achieved by our ReTanh gates but not by binary gates.
>
> >The interpretability is not quantitatively supported
>
> Thank you very much for this excellent suggestion. We ran an additional analysis in the Robot Remote Control setup. Here we took the fully trained networks (sec. 6.2, 10k epochs, 20 random seeds) and fed in all sequences of the generalization set. Each input was classified as triggering the control of the robot (control triggered) or not (no control triggered). For each input, we analyzed whether one of GateL0RD’s gates opened (any gate open) or not (no gate open). Below we report the mean gate openings for control and no control events (+- denotes standard deviation).
>
> - Hits: P(any gate open| control triggered) = 0.978 +- 0.016
> - Misses: P(no gate open| control triggered) = 0.022  +-  0.016
> - False alarms: P(any gate open| no control triggered) = 0.089 +- 0.037
> - Correct rejections: P(no gate open| no control triggered) = 0.911  +-  0.037
>
> Thus, GateL0RD seems to mostly update its latent state when the unobservable state of the environment changes (control of the robot is triggered). We thank the reviewer for the great suggestion and we will add this analysis to Suppl D.4. (Robot Remote Control: Learned latent states).
>
> > L270 "only encodes unobservable information within its latent state" this claim has, at best, anecdotal evidence (Fig 4d)
>
> We agree and we will soften the claim: “Instead, GateL0RD’s better performance is likely because it mostly encodes unobservable information within its latent state. This is shown exemplarily in Fig. 4d (bottom row) and this is analyzed in Suppl. D.4.”
>
> >Fig 3c is too zoomed-out.
>
> Thanks for pointing this out. We will modify the figure.
>
> > Fig 15 is referenced in the text but appears in the supplement.
>
> Thank you very much for noticing this. This was a mistake and we intended to reference Fig. 6c.
>
>
> References:
>
> [1] Jan Koutnik, Klaus Greff, Faustino Gomez, and Juergen Schmidhuber (2014). A clockwork RNN. In Proceedings of the 31st International Conference on Machine Learning, ICML.

---

> > ### Comment · Reviewer_sDuQ · 2021-08-28
> > **Re: response**
> >
> > These responses have addressed all of my concerns, so I will increase my rating.
> >
> > It is quite interesting that the gate opening is strongly correlated with the control triggering. However, can the classifier be reversed, to compute each of the four P({control triggered, no control triggered} | {any gate open, no gate open})? This treats the gate opening (always available) as feature for the classification of the control triggering (not available at test time, in theory).

---

### Official Review · Reviewer_yJcq · 2021-07-15

**Rating:** 6
**Confidence:** 4

**Summary:**

This work proposes an RNN with a gating mechanism that encourages its latent state to have sparse updates by incorporating stochastic gating units trained using an approximation of L0 regularization of the state updates. The authors propose to use this RNN in partially observable problems in which the latent dynamics are known to change slowly and long-term memory is important. The authors show in a number of partially observed tasks that their proposed RNN and training objective outpeforms well known RNN architectures when predicting the environment dynamics.

**Limitations And Societal Impact:**

The authors have been fairly clear about the limitations and trade-offs of their method; and as a paper focused on methodology, it does not seem to have any direct social and ethical implication.

**Main Review:**

* Originality and significance

Slow changing dynamics is a pervasive property of many time series and (PO)MDPs.  While designing architectures and training algorithms that reflects this property is not novel, it still remains an open research problem. The method proposed, an RNN that closely resembles LSTMs, is very closely related to some existing techniques as stated in the related work, with the main difference being in how the new gating units are modelled and trained. The authors note that these prior works have not been applied to POMDP problems but instead to conventional classification and language-based tasks. However, because the authors do not compare with these works, it's unclear how those techniques (see references [27, 28, 29]) would fair in POMDPs problems. The authors claim that a GRU does not include an output that is not its state (making it difficult to predict changing observations when the state remains constant), however that can be simply remedied by using the observation in addition to the state in the f_post component.

Evaluating these ideas in POMDPs is original, and the experiment section show a clear advantage of the RNNs sparsely changing bias over the baselines which is definitely interesting. Since the main novelty is the application to POMDP problems (and not as much the method itself), the selection of environments should be more extensive. The environments' partial observability and dynamics are fairly simple (e.g. they do not incorporate stochastic transitions). This paper would thus be strengthed if the authors compare relevant work in modelling the dynamics in challenging POMDPs and discussed the method's advantages and limitations (see Igl et al. 2018, Hafner et al. 2019 for variational methods for example).  If the main contribution of this work is to use the proposed RNN for long-term memory of unchanged environment states, then perhaps the paper would benefit from presenting it as a general module that can replace the core RNN in more complex architectures. For instance, many variational methods use an LSTM as the memory back-bone. Would their performance improve with this?

* Quality

The proposed RNN is tested in terms of next step prediction as well as its usefulness for planning. The second aspect is important to assess the quality of the predictions when unrolled over time.
The experiments succesfully show the benefits of sparse updates in different environments and settings (e.g. with generalization to unseen trajectories), as well as show the explainability of the learned states. There's also a good discussion on the trade-offs of predictive performance and explainability (sparseness) which is important for the reader to know. Results also seem robust as the error bars are relatively small.

The L0 regularization method is well derived and closely related to [16]. However this work, contrary to the relaxation of the intractable L0 regularization, does not minimize the probability of non-zero activations as in [16] (which is differentiable) as a proxy of L0, but instead directly incorporates a non-differentiable term based on the sum of gate values, which is approximated with the straight-through estimator. It's unclear why the authors choose this particular solution which is more biased (due to using the straight-through estimators in *both* terms of the loss), and also what practical consequences this has.
I would also like to see an ablation study where a simpler version of this idea is tried: a sigmoid gate is used with L1 regularization. In other words, to what extent is it necessary to use stochastic gates?

* Clarity

The paper is well-written and the methods are generally clear, well explained and motivated.  There are some parts where certain details are explained in the appendix but not referenced from the paper so it seemed at first a bit cryptic. For instance, in line 93 they refer to a "variation of the straight-through estimator" without specifying which variation. Similarly, many interesting ablation studies are in the appendix without any reference to them.

Some other minor issues:
- typo in line 75 (combine > combined).
- line 248, I believe the authors mean that a higher value results in *more* latent state changes?

References:

Maximilian Igl, et al., Deep Variational Reinforcement Learning for POMDPs. ICML 2018

Danijar Hafner et al., Learning Latent Dynamics for Planning from Pixels. ICML 2019

**Time Spent Reviewing:**

8

---

> ### Author Response · Authors · 2021-08-10
> **Answer to yJcq**
>
> Thank you for your time reviewing our work, suggestions for additional experiments, and opportunities for clarification.
>
> > If the main contribution of this work is to use the proposed RNN for long-term memory of unchanged environment states, then perhaps the paper would benefit from presenting it as a general module that can replace the core RNN in more complex architectures. For instance, many variational methods use an LSTM as the memory back-bone. Would their performance improve with this?
>
> Thank you for this excellent suggestion. However, our impression is that the DeepMind control suite problems considered in [1] and the flickering Atari problems of [2] are not well suited to showcase the advantages of using GateL0RD compared to other RNNs, i.e., better long-term memorization and out-of-distribution generalization.
>
> Instead, we chose to *evaluate our system in the MiniGrid environment* [3], where memorization is crucial for many different tasks. We took an existing Reinforcement learning system [4,5] and replaced the LSTM memory module with GateL0RD. We report the full results in our “General answer”. In various environments, the system with GateL0RD as a memory module outperforms the standard architecture (LSTM) in terms of mean rewards.
>
> >  I would also like to see an ablation study where a simpler version of this idea is tried: a sigmoid gate is used with L1 regularization.
>
> Thank you for this great suggestion. We ran an additional ablation study in the Robot Remote Control setting where we replaced GateL0RD's ReTanh gate with a sigmoid gate and instead minimized the $L_1$ and $L_2$ norm of gate activations. We report the results in our “General answer”. Similar to the other RNN baselines, the $L_1$ and $L_2$-variants achieve a low testing error but a high prediction error on the generalization set. This suggests that they also strongly overfit to spurious temporal dependencies, unlike our $L_0$ version.
>
> > To what extent is it necessary to use stochastic gates?
>
> We agree that this was not clear from the main text. We provided an ablation study on the effect of using stochastic vs. deterministic gates in the Supplementary Material C.3. The stochasticity of the gates seems to have a regularizing effect, resulting in fewer latent state changes while achieving the same level of prediction accuracy. We will add a sentence referring to this ablation study when introducing the stochastic gates (p. 4, right after Eq. 10): ‘The effect of gate stochasticity is analyzed in Suppl. C.3’.
>
> > line 248, I believe the authors mean that a higher value results in more latent state changes?
>
> The hyperparameter $\lambda$ scales the punishment of opening a gate (Eq. 5) and open gates are required in order to change the latent states. Thus, a higher $\lambda$ results in fewer latent state changes. We will change the sentence to: 'a higher value results in fewer gates openings and, thus, fewer latent state changes.' To further clarify this aspect, we will change the title of Fig. 3(d) to 'Number of gate openings'. We hope this eliminates any confusion.
>
> Thank you for catching the typo.
>
> References:
>
> [1] Danijar Hafner, Timothy Lillicrap, Ian Fischer, Ruben Villegas, David Ha, Honglak Lee, James Davidson (2019). Learning latent dynamics for planning from pixels. In International Conference on Machine Learning, ICML.
>
> [2] Maximilian Igl, Luisa Zintgraf, Tuan Anh Le, Frank Wood, and Shimon Whiteson.  Deep variational reinforcement learning for POMDPs. In International Conference on Machine Learning, ICML.
>
> [3] Maxime Chevalier-Boisvert, Lucas Willems, and Suman Pal (2018). Minimalistic Gridworld Environment for OpenAI Gym. https://github.com/maximecb/gym-minigrid
>
> [4] Maxime Chevalier-Boisvert, Dzmitry Bahdanau, Salem Lahlou, Lucas Willems, Chitwan Saharia, Thien Huu Nguyen, and Yoshua Bengio (2019). BabyAI: First Steps Towards Grounded Language Learning With a Human In the Loop. International Conference on Learning Representations, ICLR.
>
> [5] Lucas Willems (2018), RL Starter Files, https://github.com/lcswillems/rl-starter-files

---

> > ### Comment · Reviewer_yJcq · 2021-08-27
> > **Reply to authors**
> >
> > I appreciate the author's detailed responses and clarifications. A common concern among reviewers has been the lack of quantiative evidence for some of the claims. I believe the paper has been clearly strengthened with the additional experiments using MiniGrid, and the ablation study using sigmoid gates with L1 and L2 regularization. These new results seem to show the benefits of GateL0rd.
> > While I would still like to see additional insights (whether by derivation, or empirically) about the effects of the approximations used for the gradients of the gates (cf. ReLU gradient) and L0 regularization, I don't think these are absolutely necessary.
> >
> > That said, I still share the same concern with reviewer sDuQ regarding the minor incremental contribution of the proposed method relative to related work that propose binarized gates along with state update regularization. While the authors show experiments comparing the benefits vs purely binary gates, the related work methods [27, 28, 29] also involve other differences (for instance, in the use of Gumbel-Softmax [29], or in how the RNN is allowed to selectively gate each dimension [29]) which are not compared.
> >
> > Given the author's responses and additional expeirments I have revised my score to a marginal rejection, but my concerns above still remain.

---

> > > ### Author Response · Authors · 2021-08-29
> > > **On our contribution**
> > >
> > > Thank you for your response and for raising your score. We agree that the new experiments strengthened our submission and we would like to thank the reviewer again for the great suggestions. However, we respectfully disagree with the reviewer's concerns about the contribution of our work, which we would like to address in the following.
> > >
> > > We agree that GateL0RD is very related to previous approaches using binary gates [1, 2]. However, we believe our ablations using binary gates (Supp. C.1) demonstrate that GateL0RD and its ReTanh gates is better suited for regression problems in POMDPs. The Gumbel-Softmax gates [1] pose a different approach to binarization than the straight-through estimator. However, the Gumbel-Softmax gates still suffer from problems of binary gating, that is, they cannot interpolate between latent states and instead either replace the latent state or leave it unaltered. Selective-Activation RNNs [2] can selectively gate each dimension. However, so can GateL0RD. Because the ReTanh gate $\Lambda$ is applied for each dimension of its input $s^i_t$, it can selectively update each dimension individually. For example, in the Billiard Ball setting, as exemplary shown in Fig. 6 (b), GateL0RD only updates two dimensions over the course of the sequence.
> > > Supp D.1 analyzes the selectivity of the gating for the Billiard Ball scenario in more detail. In sum, we believe that our ablation demonstrates that our novel ReTanh gate is better suited for regression problems than the binary gates of those related approaches.
> > >
> > > However, unlike the other approaches that use binary gates [1, 2], we are, to the best of our knowledge, the first to apply sparsely gated latent state updates to prediction and control in POMDPs. Including the new MiniGrid experiments, we show improvements not only for model-predictive control but also for reinforcement learning. We demonstrate in which cases such sparse latent updates are useful (generalization from training scheme, spurious temporal dependencies, long-term memory, sample efficiency, and explainability), using overall 9 problems. Thus, we believe these experimental investigations alone are an interesting contribution for prediction and control in POMDPs.
> > >
> > >
> > > References:
> > >
> > > [1] Zhuohan Li, Di He, Fei Tian, Wei Chen, Tao Qin, Liwei Wang, and Tieyan Liu (2018). Towards binary-valued gates for robust LSTM training. Proceedings of the 35th International Conference on Machine Learning
> > >
> > >
> > > [2] Thomas Hartvigsen, Cansu Sen, Xiangnan Kong, and Elke Rundensteiner (2020). Learning to selectively update state neurons in recurrent networks. Proceedings of the 29th ACM International Conference on Information & Knowledge Management

---

> > > > ### Comment · Reviewer_yJcq · 2021-09-01
> > > > **On the paper's contributions**
> > > >
> > > > Dear authors,
> > > > I appreciate your detailed answers to my concerns. Since the main text contains only a brief section dedicated to discussing the most related methods, it seemed to me that insufficient discussion and comparisons where provided. After our discussions here and your further clarifications, as well as the ablation studies in the supp. material, many questions a reader might have are answered. I do think your contribution has useful benefits over the mentioned methods, and is especially interesting when taking into account the focus on POMDPs.
> > > > While the method itself is of somewhat limited technical novelty, for the reasons above I will revise my score to a weak accept. The paper will be strengthened if the authors incorporate some of the discussion above into the related work section and include references to the ablation studies where possible.

---

### Official Review · Reviewer_EaBh · 2021-07-16

**Rating:** 6
**Confidence:** 4

**Summary:**

This paper proposes a new recurrent neural network architecture (RNN) called GateL0RD, designed to learn sparsely changing latent variables of sequential data. The authors hypothesize that latent variables should be updated sparingly only if significant changes happen in the considered environment. To enforce this hypothesis in RNN, GateL0RD uses a new gating function of Rectified Tanh to output exactly zero values. For actual implementation, the considered prediction task is regularized to minimize the number of positive elements in the sampled gate inputs. Ablation study shows that GateL0RD improved prediction accuracy and planning performance compared with LSTM or GRU in several POMDP environments.

**Limitations And Societal Impact:**

Yes, the authors have addressed the limitations and societal impact.

**Main Review:**

(1) Originality \
This work builds a new activation function (ReTanh) on a gating function in usual LSTM/GRU architectures where sigmoid and $\tanh$ are widely used.

(2) Quality
1) Simple but practical implementation of $L_0$ regularization makes this work scalable.
2) Various experimental results and the corresponding ablation study are provided, including the interpretability.
3) Main concern of this work is the support of the hypothesis that latent variables should change sparsely. It looks like there is no theoretical support or guarantee on the optimality of this approach, though several examples in the real-world and human recognition area are mentioned.

Q1. Is there any more substantial explanation or related work supporting the effectiveness of the hypothesis? \
Q2. Does the proposed method perform better than $L_1$, $L_2$ regularization methods?

(3) Clarity \
This paper is well written and easy to read. The strong points of this work are well organized regarding items: accuracy, generalization, long-term memorization, and interpretability.

(4) Significance \
As mention in the Discussion section, the proposed regularization may not be effective when the considered partial domain has severely changing characteristics.

Q3. Is there any backup experiment (any specific environment) to check this aspect? Can this regularization be used in general in partially observable domains? Setting $\lambda=0$ and asserting that "the unregularized network performs well." is not persuasive. The proposed architecture without the regularization is not fundamentally different from LSTM / GRU. Also, note that we usually do not know a priori how much the considered environment changes.


**Time Spent Reviewing:**

12

---

> ### Author Response · Authors · 2021-08-10
> **Answer to EaBh**
>
> Thank you for your time reviewing our work and the very useful feedback.
>
> > Q1. Is there any more substantial explanation or related work supporting the effectiveness of the hypothesis?
>
> Our work was mainly inspired by the observations of properties of the physical world, which we outlined in the introduction, and theories of human event cognition. However, we realized that there is also a link to causality research. Thus, we will add a paragraph to our introduction (p.2 l.48). The main argument is that models that capture the causal dependence or independence [1,2,3] enjoy improved generalization. Updating the latent states every time-step contradicts with this goal, because doing so creates dependencies of the latent state not only on the current input of the system, but also on all previous inputs. Thus, by suitably segmenting the dependencies of latent variables over time one can expect improved generalization across spurious temporal correlations.
>
> > Q2. Does the proposed method perform better than $L_1$ or $L_2$ regularization methods?
>
> Thank you for this great question. In a new ablation experiment we compared against $L_1$ and $L_2$ regularization. We report the results in our general answer. Both ablated systems achieve a very small prediction error on the testing set but have a large error on the generalization set, similar to standard GRUs and LSTMs.
>
> > Q3: Is there any backup experiment (any specific environment) to check this aspect?
>
> This aspect becomes visible in our additional experiment in the Supplementary Material D.5. Here we use our modified version of the Fetch Pick&Place environment in which all velocities are removed from the observations. Because in Fetch Pick&Place position control of the end-effector is realized by a PID-controller running at a higher frequency, continuous latent state updates are advantageous for predicting the positions. As shown in Figure 18, stronger regularizations (higher $\lambda$) of GateL0RD result in larger testing errors.
>
> > Setting $\lambda=0$ and asserting that "the unregularized network performs well." is not persuasive
>
> Thank you for bringing this up. We modified the sentence to: “As demonstrated by an additional experiment in Suppl. D.5., the unregularized network performs well in such cases.”. We also think that we missed an opportunity to highlight the advantages of our network without regularization ($\lambda=0$):
>
> “As shown in Fig. 3d, even without regularization ($\lambda = 0$) GateL0RD continuously decreases the mean number of gate openings. After 5k epochs, GateL0RD on average opens a gate less than 50% of the time. This effect emerges from the interplay of stochastic gradient descent and the ReTanh having gradients of 0 for inputs $s_t \leq 0$. Over training time, gates will randomly close and kept shut if they do not contribute to decreasing the loss. This effect is closely related to the “dying ReLU problem” when using ReLU activation functions [4]. While dying ReLUs are considered a problem, in our case this is advantageous whenever the gate regularization is beneficial. We believe that this results in GateL0RD with $\lambda = 0$ being more robust to out-of-distribution shifts than GRUs and LSTMs. For example GateL0RD with $\lambda = 0$ achieves a smaller mean autoregressive prediction error when trained using teacher forcing (Fig 3a), compared to the baseline RNNs.”
>
> In response to your review we are adding the paragraph above to our supplementary section D.1 (Billiard Ball: Analyzing latent states). Additionally, we are adding two sentences to the last paragraph of section 61.: ‘Note that even without regularization ($\lambda = 0$) GateL0RD learns to use fewer gates over time. We describe this effect in more detail in Suppl. D.1.’ We hope this additional information and analysis improves our system description.
>
>
> References:
> [1] Jonas Peters, Dominik Janzing, and Bernhard Schölkopf (2017). Elements of causal inference: foundations and learning algorithms. The MIT Press.
>
> [2] Bernhard Schölkopf (2019). Causality for machine learning. arXiv preprint arXiv:1911.10500.
>
> [3] Bernhard Schölkopf, Francesco Locatello, Stefan Bauer, Nan Rosemary Ke, Nal Kalchbrenner, Anirudh Goyal, and Yoshua Bengio (2021). Towards Causal Representation Learning. Proceedings of the IEEE, 109(5), 612-634.
>
> [4] Lu Lu, Yeonjong Shin,Yanhui Su, and George Em Karniadakis (2019). Dying relu and initialization: Theory and numerical examples. arXiv preprint arXiv:1903.06733.

---

> > ### Comment · Reviewer_EaBh · 2021-08-23
> > **Additional Question on New Experiments**
> >
> > Dear Authors
> >
> > Thank you for the detailed explanations of my comments and new experiments. I have additional questions regarding the investigations in Minigrid.
> >
> > In the code ‘RL Starter Files’ you used, a key hyperparameter called **recurrence** is the step length of the gradient backpropagation (or the length of memory, briefly speaking). For RedBlueDoors-6x6, for example,
> >
> > python3 -m scripts.train --algo ppo --env MiniGrid-RedBlueDoors-6x6-v0 --model RedBlueDoors --**recurrence** 4 --save-interval 10 --frames 1000000,
> > as shown on the website.
> >
> > Q1. For LSTM, what are the values of the **recurrence** for the five environments you used? Did you tune this value for each of the five environments (or use the same value for simplicity, e.g., 4)? If you did tuning, did the larger recurrence value produce better mean rewards?
> >
> > Q2. For GateL0RD, what are the values of the **recurrence** for the five environments you used? Did you (a) tune this value for each of the five environments, or (b) use the same value used in the LSTM case? (I guess you probably used (b). If you used (a), did the larger recurrence value produce better mean rewards?) Do you think your choice between (a) and (b) is the fairer comparison than the other?
> >
> > For clarity, did you use higher-dimensional (7x7x3 grids) default input or RGB version?
> >
> > Thanks again for your response.

---

> > > ### Author Response · Authors · 2021-08-23
> > > **Minigrid Experiment Details**
> > >
> > > Thank you for your question.
> > >
> > >
> > > Q1. For consistency, we used the same hyperparameters across all environments. Whenever possible, we set the hyperparameters to the default values from [1] and used the PPO parameters from the original paper [2].
> > > However, for some of the environments we considered, e.g., MemoryS13Random, dependencies over longer time horizons need to be learned in order to solve the task.
> > > Thus, __recurrence__ is one of the few hyperparameters that we modified from the default value. In fact, the value of 4 was only an example in the code [1] and in the paper [2] it is 20 (or 80 only for big environments).
> > > We set the __recurrence__ hyperparameter to 32, as it works well for all environments for the LSTM baseline.
> > >
> > >
> > > Q2. One of the goals of the MiniGrid experiments was to examine if GateL0RD can improve existing frameworks for POMDPs if it replaces the internal RNN.
> > > Therefore, we did not additionally tune the hyperparameters when using GateL0RD.
> > > Instead, we took the hyperparameter setting that worked well for the standard architecture (LSTM) and simply replaced the LSTM with GateL0RD with a suitable $\lambda$ ($\lambda = 0.01$).
> > > We think this is a fair comparison if not advantageous for the LSTM-architecture since an even better performance could be expected for the GateL0RD-architecture when optimizing the hyperparameters specifically for GateL0RD.
> > >
> > >
> > > > For clarity, did you use higher-dimensional (7x7x3 grids) default input or RGB version?
> > >
> > >
> > > We used the higher-dimensional default input (7x7x3 grids).
> > >
> > >
> > > References:
> > >
> > > [1] Lucas Willems (2018), RL Starter Files, https://github.com/lcswillems/rl-starter-files
> > >
> > >
> > > [2] Maxime Chevalier-Boisvert, Lucas Willems, and Suman Pal (2018). Minimalistic Gridworld Environment for OpenAI Gym. https://github.com/maximecb/gym-minigrid

---

> > > > ### Comment · Reviewer_EaBh · 2021-09-01
> > > > **Reply**
> > > >
> > > > Thanks for all the comments. I carefully read the authors' responses and revised my rating accordingly.

---

### Official Review · Reviewer_qktj · 2021-07-19

**Rating:** 7
**Confidence:** 4

**Summary:**

The manuscript presents a novel recurrent neural network architecture with an inductive bias for learning sparsely changing hidden state representations. This seems to be achieved by the proposed gating scheme using the novel rectified tanh activation function for the update gate values. This activation function yields true zero activations producing an exact copy of the corresponding value from the previous hidden state. By applying L0 regularization to the state-updates, the network is incentivized to minimize the number of modified dimensions in the state.

**Ethical Concerns:**

None.

**Limitations And Societal Impact:**

See main review.

**Main Review:**

The presentation is very clear and detailed. The experiments highlight several contributions: 1) GateL0rd develops stable, sparsely updated latent state representations. 2) making the output also depend directly on the input, frees the hidden state to focus on information inferred from past time steps. To summarize the points made in the sections below, I think the work proposes an intriguing alternative to commonly used RNN architectures, but the experimental evaluation has some weaknesses unless I’m misinterpreting some observations.

Strengths
1. The proposed regularized ReTanh gating and direct dependence of the network output on the input seem like very appropriate solutions to weaknesses of other (gated) RNN architectures. Especially the potential disentanglement of observable and unobservable information might help in reinforcement learning with an unknown reward function.
1. The sparse state changes should be very useful for event-based sequence modeling.
1. GateL0rd seems to outperform other commonly used RNN architectures.
1. The manuscript is very well written. A lot of questions that came up while reading were answered either a few paragraphs later or in the supplementary material. For instance, I was a bit sceptical at first about the choice of the ReLU gradient estimator, but Section C.2 cleared that up for me.

Limitations/Questions
1. How do you think GateL0rd would perform if applied to environments with very high-dimensional visual observations, such as MuJoCo, Atari or ProcGen (of course using an appropriate encoder network for the frames)? I think that GateL0rd certainly has a better prior for disentangling factors of variation in the environment than GRUs or LSTMs. However, I also think that it may be difficult to learn this disentanglement, if at the same time a compatible input representation has to be discovered given high-dimensional observations.
1. Footnote 3: I think that tanh(s/2) = 2 * sigmoid(s) - 1.
1. Can you quantify the impact of $f_{init}(x)$ compared to setting $h_0=0$?
1. The testing error curves in Figures 4 and 5 look like training did not  yet converge. If that is the case, the networks should be trained longer since the focus of the manuscript does not seem to be on comparing the sample efficiency of different architectures.
If training did converge, I’d suggest presenting the loss curves as well.
1. Can the test prediction error going up for GRU and LSTM during training really be attributed to the networks’ overfitting to observable information and spurious correlations? The variance of these curves seems to be very large which is often a symptom of optimization issues (e.g. exploding gradients for some runs)!
1. In the shepherd task, suppose the sheep was allowed to move horizontally while occluded with horizontal motions sampled from some symmetric zero-mean distribution. If the motions were part of the observation, do you think GateL0rd would be able to learn to modify the memorized initial position accordingly in its hidden state? If the motion information were not observed, how would GateL0rd learn to account for the uncertainty?
1. Time indices are not consistent. In line 106 the mapping is (y_t, h_{t+1}) = f(x_t, h_t), while in line 71 it is (y_t, h_t) = f(x_t, h_{t-1}). Also in Equation 2, I believe the regularization should be applied to delta h_t.


**Time Spent Reviewing:**

2

---

> ### Author Response · Authors · 2021-08-10
> **Answer to qktj**
>
> Thank you for your time and excellent feedback. We are very happy that you
> found our representation clear and well written.
>
> > How do you think GateL0rd would perform if applied to environments with very high-dimensional visual observations?
>
> We thank the reviewer for this great suggestion. We ran additional experiments to test GateL0RD’s performance in a reinforcement learning setting in the MiniGrid simulator where the network uses convolution to preprocess an image-like input. Please check the “General Answer” for details.
>
> > Can you quantify the impact of $f_{\text{init}}$  compared to setting it to 0?
>
> Thank you for pointing out the lack of this ablation. We ran a new ablation experiment in the Billiard Ball setting trained with scheduled sampling (as in sec. 6.1.) to quantify the impact of $f_{\text{init}}$. Below we report the mean (+- std. dev) test prediction error (over 20 random seeds) after full training (the lower the better):
>
> Billiard Ball:
> ```
>        | GateL0RD            | LSTM               | GRU                | Elman RNN
> f-init | 0.00827 +- 0.00050  | 0.00955 +- 0.00076 | 0.00908 +- 0.00052 | 0.01439 +- 0.00206
> h0=0   | 0.02633 +- 0.015351 | 0.02382 +- 0.01114 | 0.01199 +- 0.00061 | 0.03487 +- 0.00684
> ```
>
> All networks achieve a smaller prediction error with  $f_{\text{init}}$. The effect seems to also depend on the type of network used. GRUs suffer less from setting $h_0=0$ than the other networks. We will add the results to the final paper.
>
> > The testing error curves in Figures 4 and 5 look like training did not yet converge.
>
> We now train the Robot Remote Control environment for 10k training steps. No qualitative differences appear. We will extend the plots and add the loss-curves to the supplementary.
>
> > Can the test prediction error going up for GRU and LSTM during training really be attributed to the networks’ overfitting to observable information and spurious correlations?
>
> The standard deviations in this task are indeed rather high. We determined the learning rate using a grid search and took the value with the lowest mean error over two random seeds thereby ignoring standard deviations. However, as shown in the Supplementary Material D.2 even with a lower learning rate optimized for generalization the baseline RNNs achieve a much worse generalization error. Taken together, we believe these two findings suggest that the baseline RNNs really overfit to the training data. Since the training data differs from the generalization data in the temporal correlations of the actions, we believe our statement is valid.
>
> > In the shepherd task, suppose the sheep was allowed to move horizontally while occluded with horizontal motions sampled from some symmetric zero-mean distribution. If the motions were part of the observation, do you think GateL0rd would be able to learn to modify the memorized initial position accordingly in its hidden state? If the motion information were not observed, how would GateL0rd learn to account for the uncertainty?
>
> If needed for prediction, GateL0RD will adapt its hidden variables accordingly; potentially a smaller value of $\lambda$  is required. For the unobserved motions, GateL0RD could be adapted to output distribution parameters, as in Gaussian Neural Networks[5], and can be trained with NLL loss.
>
> Thank you very much for the careful reading and for catching some mistakes and inconsistencies. We fixed them.
>
> References:
>
> [1]  Maxime Chevalier-Boisvert, Lucas Willems, and Suman Pal (2018). Minimalistic Gridworld Environment for OpenAI Gym. https://github.com/maximecb/gym-minigrid
>
> [2] Maxime Chevalier-Boisvert, Dzmitry Bahdanau, Salem Lahlou, Lucas Willems, Chitwan Saharia, Thien Huu Nguyen, and Yoshua Bengio (2019). BabyAI: First Steps Towards Grounded Language Learning With a Human In the Loop. International Conference on Learning Representations. ICLR
>
> [3] John Schulman, Filip Wolski, Prafulla Dhariwal, Alec Radford, and Oleg Klimov (2017). Proximal policy optimization algorithms. arXiv preprint arXiv:1707.06347.
>
> [4] Lucas Willems (2018), RL Starter Files, https://github.com/lcswillems/rl-starter-files
>
> [5] Christopher M Bishop (2006). Pattern recognition and machine learning. Springer

---

> > ### Comment · Reviewer_qktj · 2021-08-25
> > **thanks for response.**
> >
> > I have carefully read author response and other reviews, given the other reviewers valid points I stay with my rating.

---

### Author Response · Authors · 2021-08-10
**General Answer**

We thank the reviewers for their thorough and helpful reviews. Since the reviewers liked our paper and had mainly concerns about the empirical evidence we decided to use their constructive feedback and conducted several additional experiments to support our claims.

In summary, we compared our $L_0$ regularization against $L_1$ and $L_2$ regularized versions, ran ablation experiments to analyze the effect of $f_\text{init}$ (see answer to reviewer qktj), compared to Clockwork RNNs (see answer to reviewer sDuQ), and analyzed the gate usage quantitatively (see answer to reviewer sDuQ). Most importantly, we applied GateL0RD in a Reinforcement Learning setting in the MiniGrid gridworld domain [1]. We present the results in the following and will include their description and result in the final version of the paper.

##  $L_1$ and $L_2$  (EaBh & yJcq)
We ran an additional ablation study where we replaced GateL0RD's ReTanh gate with $L_0$ regularization with a sigmoid gate and instead minimized the $L_1$ and $L_2$ norm of gate activations as suggested. We tested this ablation in the Robot Remote Control experiment (as in section 6.2.) with an appropriate value of $\lambda$ for each. Results: final mean (+-std.dev) prediction errors after 10k epochs over 20 random seeds.

```
               |  GateL0RD (L0)     |    L1 Ablation     |    L2 Ablation
test set       | 0.00015 +- 0.00010 | 0.00014 +- 0.00011 | 0.00008 +- 0.00004
generalization | 0.00110 +- 0.00071 | 0.02091 +- 0.00202 | 0.02170 +- 0.00152
```

Both ablations behave similarly to LSTMs or GRUs: While they achieve a very small prediction error on the testing set, their prediction error on the generalization set is large. This supports the necessity of our $L_0$ regularized ReTanh gate.

## Higher-dimensional input and downstream tasks - MiniGrid (qktj & yJcq)

We ran a series of additional experiments in the MiniGrid environment [1], a library of partially observable benchmark reinforcement learning problems in grid worlds. Here the input is a higher-dimensional (7x7x3 grids) egocentric view of the agent on its environment.

We build upon previous research to solve the tasks in MiniGrid. Chevalier-Boisvert et al. [2] used PPO for learning a policy. The network architecture has a convolutional network for preprocessing the input (akin to $f_{pre}$ in Fig. 1d), a subsequent LSTM as a memory module (akin to $f_{\theta}$ in Fig. 1d), and two subsequent MLPs (similar to $f_{post}$ in Fig. 1d) for policy and value estimations. We start from an implementation by one of the authors [3] and replaced the LSTM with GateL0RD ($\lambda = 0.01$) as a simple drop-in replacement and we left the other hyperparameters unmodified.
Below we report the mean (+- std.dev.) rewards obtained (10 random seeds) over the course of training for various environments that require memory (see [1] for details):

### RedBlueDoors-6x6
```
frames(1e5) | 3              | 6              | 9               | 12             | 15
LSTM        | 0.084 +- 0.041 | 0.587 +- 0.266 | 0.926 +- 0.042  | 0.957 +- 0.008 | 0.961 +- 0.003
GateL0RD    | 0.197 +- 0.109 | 0.866 +- 0.173 | 0.954 +- 0.0182 | 0.960 +- 0.006 | 0.961 +- 0.003
```

### SimpleCrossingS9N3
```
frames(1e5) | 5              | 10             | 15             | 20             | 25
LSTM        | 0.148 +-0.061  | 0.463 +-0.124  | 0.791 +-0.079  | 0.862 +- 0.029 | 0.890 +- 0.015
GateL0RD    | 0.320 +- 0.155 | 0.768 +- 0.082 | 0.857 +- 0.044 | 0.865 +- 0.027 | 0.891 +- 0.012
```

### DoorKey-8x8
```
frames(1e5) | 10             | 20            | 30             | 40            | 50
LSTM        | 0.024 +-0.017 | 0.057 +-0.027 | 0.671 +-0.347  | 0.950 +-0.004 | 0.951 +-0.002
GateL0RD    | 0.031 +-0.020 | 0.446 +-0.416 | 0.856 +- 0.271 | 0.951 +-0.003 | 0.953 +-0.003
```

### KeyCorridorS3R2
```
frames(1e6) | 10            | 20            | 30            | 40             | 50
LSTM        | 0.007 +-0.006 | 0.141 +-0.259 | 0.714 +-0.335 | 0.798 +- 0.264 | 0.798 +-0.264
GateL0RD    | 0.012 +-0.011 | 0.727 +-0.307 | 0.880 +-0.004 | 0.881 +- 0.004 | 0.882 +-0.005
```

### MemoryS13Random
```
frames(1e6) | 10             | 20            | 30             | 40            | 50
LSTM        | 0.483 +- 0.063 | 0.768 +-0.136 | 0.894 +-0.124  | 0.938 +-0.074 | 0.960 +-0.063
GateL0RD    | 0.500 +- 0.046 | 0.893 +-0.164 | 0.979 +- 0.003 | 0.980 +-0.004 | 0.979 +-0.004
```

The architecture containing GateL0RD manages to outperform the standard architecture (with LSTM) in terms of mean rewards in all environments. In particular, GateL0RD is much more sample efficient reliably achieving a high reward with less training than the unmodified architecture. We will include this new experiment in the final version of the paper.

[1]  M. Chevalier-Boisvert, L. Willems, and S. Pal (2018). Minimalistic Gridworld Environment for OpenAI Gym. https://github.com/maximecb/gym-minigrid

[2] M. Chevalier-Boisvert, D. Bahdanau, S. Lahlou, L. Willems, C, Saharia, T. H. Nguyen, and Y. Bengio (2019). BabyAI: First Steps Towards Grounded Language Learning With a Human In the Loop. ICLR.

[3] L. Willems (2018), RL Starter Files, https://github.com/lcswillems/rl-starter-files

---

### Decision · Program_Chairs · 2021-09-27

**Decision:**

Accept (Poster)

**Comment:**

The paper presents a recurrent architecture with an inductive bias (via gating and L0 regularization) that encourages sparse updates to its state. This is hypothesized to lead to improved generalization, and is empirically evaluated in a number of partially observable environments.

The reviewers were initially sceptical, but the discussion with the authors has allayed most concerns, so now the consensus is towards acceptance. In general, the reviewers agree that the paper is well written and the contribution is interesting (albeit somewhat incremental). Overall, I'm happy to recommend acceptance.